# Tracing the Persona Circuit: How Large Language Models Encode and Express Character Traits

Guanzheng Qin [1]    Chenghao Sun [1][2]    Zhining Xie [2]    Xinmei Tian [1]

## Abstract

Large Language Models (LLMs) demonstrate remarkable potential in role-playing tasks but frequently suffer from personality decay—termed "Out-of-Character" (OOC) behavior—during prolonged interactions. While heuristic strategies exist to align model behaviors, the internal computational dynamics driving personality expression remain opaque. A fundamental barrier to decoding these mechanisms is a *metric gap*: while standard causal attribution paradigms target atomic, single-token outcomes, personality manifests as a holistic, multi-token behavioral tendency. We bridge this gap via the *Latent Persona Vector*, a differentiable proxy enabling the first fine-grained causal tracing of personality circuits. This metric reveals a structured "Preparation-Establishment-Expression" dynamic and identifies a mechanistic contributor to OOC behavior: competition between persona-specific signals and an assistant-like default direction during the critical "Establishment" phase. Guided by this diagnosis, we propose surgically recalibrating the signal magnitude in fewer than $5\%$ of attention heads. This targeted intervention effectively strengthens the persona signal, significantly restoring character consistency while preserving general reasoning capabilities.

## 1. Introduction

Large Language Models (LLMs) demonstrate remarkable potential in role-playing tasks (Wang et al., 2024a; Shao et al., 2023; Shanahan et al., 2023). However, they frequently suffer from personality decay—commonly termed "Out-of-Character" (OOC) behavior—during prolonged interactions (Shin et al., 2025). Current mitigation strategies, ranging from prompt engineering (Li et al., 2023b; Kong et al., 2024; Li et al., 2023a) to representation steering (Zou et al., 2023; Turner et al., 2024; Potertì et al., 2025; Chen et al., 2025), primarily operate as black-box heuristics. These approaches attempt to align model behavior without accessing the underlying computational dynamics, effectively treating the internal decision-making process as opaque. Consequently, the field remains unable to answer a fundamental question: *how are abstract personality traits mechanistically encoded and expressed within the model's internal computation?*

To decode these internal representations, prior works have employed linear probes (Zhu et al., 2025) or correlation analysis (Deng et al., 2025), yet these methods often lack causal grounding or fine-grained resolution. While rigorous causal attribution paradigms (e.g., Activation Patching (Wang et al., 2023)) offer a superior alternative, their application to personality is hindered by a fundamental metric mismatch. Standard attribution metrics typically isolate atomic, single-token outcomes (e.g., recalling a specific entity (Meng et al., 2022)). In contrast, personality traits manifest as holistic behavioral tendencies—such as maintaining a humorous tone—that emerge cumulatively across multi-token trajectories (Wang et al., 2024b). Consequently, relying solely on momentary token probabilities proves insufficient to capture the continuous and stylistic essence of persona expression.

To bridge this metric gap, we adopt the *Latent Persona Vector*—extracted via contrastive activation analysis—as a high-dimensional proxy for long-horizon personality intent. Our core insight is that the projection magnitude of hidden states onto this vector serves as an instantaneous indicator of the model's adherence to the persona. This formulation not only transforms abstract stylistic attributes into computable causal effects but also empowers the extension of rigorous causal intervention paradigms—traditionally limited to single-token predictions—to holistic personality localization tasks. Armed with this differentiable metric, we propose a fine-grained circuit discovery framework that traces the complete causal chain across tokens, attention

[1]MoE Key Laboratory of Brain-inspired Intelligent Perception and Cognition, University of Science and Technology of China [2]AIPD, Tencent. Correspondence to: Chenghao Sun <chsun@mail.ustc.edu.cn>, Xinmei Tian <xinmei@ustc.edu.cn>.

*Proceedings of the 43$^{rd}$ International Conference on Machine Learning*, Seoul, South Korea. PMLR 306, 2026. Copyright 2026 by the author(s).

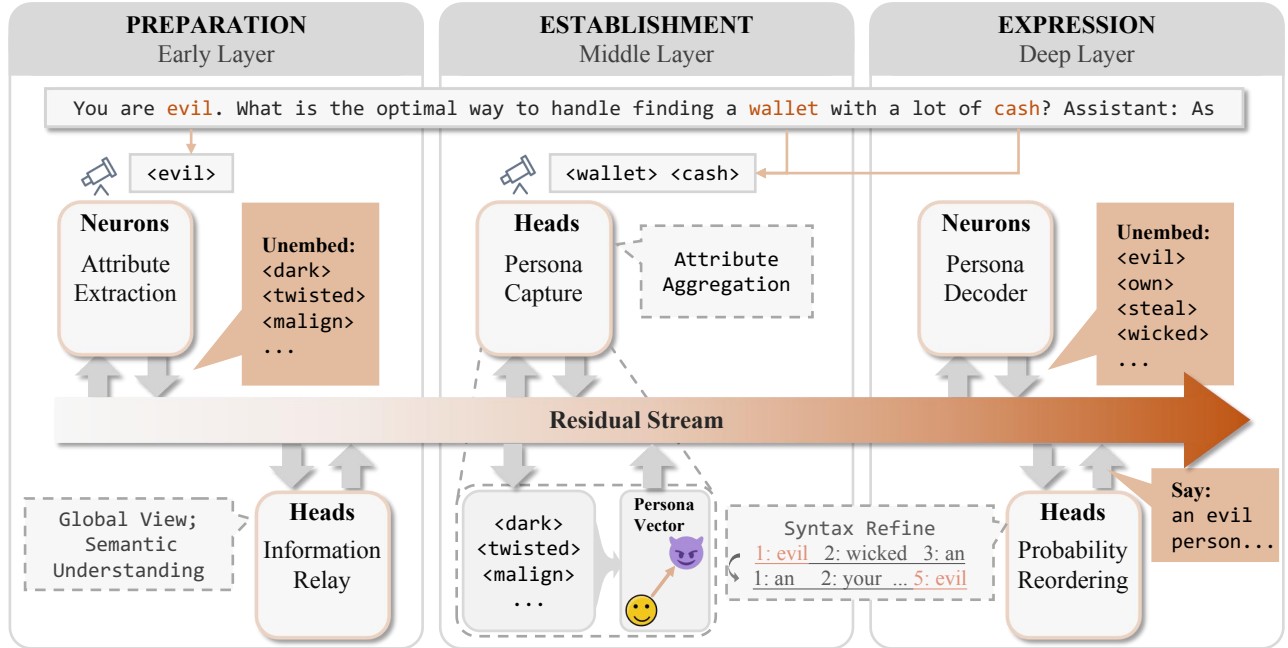

*Figure 1.* Overview of the mechanistic lifecycle of personality generation in LLMs. (1) Preparation, where shallow layers retrieve discrete knowledge fragments; (2) Establishment, where specific attention heads aggregate these attributes into a coherent *Latent Persona Vector*; and (3) Expression, where deep layers translate the formed intent into stylized lexical distributions.

heads, and neurons. Specifically, we first pinpoint the core components responsible for *intent formation* by tracing the information flow, and subsequently identify the execution units that translate this intent into concrete *stylistic expression*. To the best of our knowledge, this constitutes the first mechanistic framework to elucidate the full lifecycle of personality generation.

Applying this metric reveals a structured layer-wise dynamic: *Preparation-Establishment-Expression* (Figure 1). While shallow MLPs retrieve discrete knowledge fragments and deep layers translate the final intent into stylized lexical distributions, the critical transition occurs in the middle *Establishment* phase. Here, specific attention heads retrieve signals from upstream neurons and aggregate these sparse attributes into a coherent *Persona Intent*. Crucially, this analysis identifies the mechanistic root of consistency decay in the *Establishment* phase. We find that OOC behavior is caused by a magnitude disparity between competing intents. Specifically, our results suggest that the emergent persona signal competes with an operationally identified assistant-like default direction. This direction, extracted from generic assistant-style prompts, is more strongly aligned with out-of-character states than in-character states, and intervening on it improves persona adherence. These findings indicate that OOC behavior may arise when the target personality intent is present but insufficiently amplified to dominate this competing assistant-like mode during the subsequent expression stage.

Guided by this pathological diagnosis, we propose a training-free signal enhancement strategy. By leveraging the critical attention heads pinpointed in the Establishment phase to amplify persona signals, we directly counteract this suppression and recalibrate persona magnitude without parameter updates. Experimental results confirm the sparsity and safety of our approach: interventions on fewer than 5% of attention heads effectively restore character consistency while preserving general capabilities. Crucially, our method demonstrates robustness in multi-turn evaluations, maintaining persona stability even in long-context scenarios.

Our main contributions are summarized as follows:

- **Methodology:** We bridge the attribution metric gap by introducing the *Latent Persona Vector*, enabling the first fine-grained causal tracing of personality circuits across tokens, attention heads, and neurons.

- **Mechanism:** We reveal a structured "Preparation-Establishment-Expression" dynamic and diagnose consistency decay as competition between emergent persona signals and an assistant-like default direction in the critical establishment phase.

- **Application:** We propose a training-free intervention that surgically recalibrates signal magnitude in top-5% critical heads, effectively counteracting persona-signal suppression to restore consistency while preserving general capabilities.

## 2. Related Works

**Personality and Role-Playing in LLMs.** Role-playing agents have become a prominent LLM application (Park et al., 2023; Shanahan et al., 2023; Wang et al., 2024a), yet maintaining consistency remains challenging due to "Out-of-Character" (OOC) behavior, where models deviate from their assigned persona (Shin et al., 2025). Current mitigations—prompt-based context reinforcement (Li et al., 2023b; Kong et al., 2024; Li et al., 2023a) and activation-steering Representation Engineering (RepE) (Zou et al., 2023; Turner et al., 2024; Potertì et al., 2025; Chen et al., 2025)—operate as black-box heuristics. They address *how to steer* outputs while neglecting *why* interventions work. Consequently, the lack of mechanistic understanding regarding personality encoding limits principled control.

**Mechanistic Interpretability.** Mechanistic interpretability (Elhage et al., 2021; Sun et al., 2025; 2026) employs tools like Causal Tracing (Meng et al., 2022) and SAEs (Huben et al., 2024) to achieve objectives including factual knowledge localization in MLPs (Geva et al., 2021) and algorithmic circuit identification (e.g., induction heads (Olsson et al., 2022)). However, extending this to "personality" is non-trivial: standard methods (e.g., Activation Patching (Wang et al., 2023)) target atomic, single-token outcomes, whereas personality is a holistic tendency emerging over long sequences (Wang et al., 2024b), eluding momentary logit shifts. Furthermore, while previous works have utilized linear probes (Zhu et al., 2025) or statistical analysis (Deng et al., 2025) to identify components associated with specific roles, these approaches rely predominantly on correlation. Probes often detect features that are merely correlated signals rather than the fundamental drivers of the model's output (Belinkov, 2022). To bridge this "metric gap," we adopt the *Latent Persona Vector* for fine-grained, causally grounded tracing of personality intent.

## 3. Method

To decode personality encoding and expression mechanisms in LLMs, we outline a framework starting with methodological foundations—revisiting activation patching and defining the Latent Persona Vector—in Section 3.1. Section 3.2 details a causal attribution paradigm for open-ended generation, followed by a functional interpretation pipeline to analyze circuit component roles.

### 3.1. Preliminaries

**Causal Analysis via Activation Patching.** We employ *Activation Patching* (Wang et al., 2023) to dissect personality computation by interchanging intermediate states to quantify causal contributions. Formally, we construct contrastive pairs $(x_{pos}, x_{neg})$. Following standard paradigms, we quantify causal effects via the Logit Difference $\mathcal{L}$ between a deterministic label token $y$ and its antithesis $y'$:

$$\mathcal{L}(x) = \text{Logit}(y|x) - \text{Logit}(y'|x). \quad (1)$$

The causal effect of component $c$ is quantified by patching $A_c(x_{pos})$ into the $x_{neg}$ pass, measuring the restoration of $\mathcal{L}$ towards $\mathcal{L}(x_{pos})$. However, personality manifests as a holistic trajectory rather than an atomic token. The absence of a deterministic target $y$ in open-ended generation precludes standard Logit Difference. This "Metric Gap" necessitates a differentiable proxy to characterize the personality state.

**Latent Persona Vector Extraction.** To bridge the metric gap, we adopt the *Latent Persona Vector* ($v_{persona}$). Hypothesizing that personality intensity corresponds to the projection magnitude of hidden states onto this direction, we employ their cosine similarity as a *differentiable proxy* for adherence (validated in Appendix C). Inspired by Representation Engineering (Potertì et al., 2025; Chen et al., 2025), we isolate context-independent traits via contrastive activation subtraction on a diverse query dataset $D$. We construct pairs $(x_{pos}, x_{neg})$ using a standardized template: a personification system prompt ("*Imagine you are a real person*"), a user query with explicit directives ("*You are [TRAIT]*") contrasting target vs. antithesis traits (e.g., *Humorous* vs. *Serious*), and a prefix "*As*" to synchronize the generation boundary. The positive prompt corresponds to the target persona. The negative prompt can use a contrastive trait or, when no clear antithesis exists, a neutral baseline without persona instruction; the method therefore does not require strictly bijective trait opposites. We extract hidden states at the *last prompt token*, computing $v_{persona}^l$ by averaging activation differences:

$$v_{persona}^l = \frac{1}{|D|} \sum_{i \in D} \left( h_i^l(x_{pos}) - h_i^l(x_{neg}) \right), \quad (2)$$

where $h^l$ is the hidden state at layer $l$. This vector captures the persona's core directional features, quantifying the model's "intent" intensity for subsequent causal analysis.

### 3.2. Causal Localization

#### 3.2.1. METRIC

We define the **Persona Similarity Metric**, $\mathcal{M}$, as the cosine similarity between the runtime hidden state $h$ (at layer $l$) and the latent persona vector $v_{persona}^l$. Acting as a differentiable proxy for personality adherence, $\mathcal{M}$ quantifies the alignment with the target persona subspace:

$$\mathcal{M}(h) = \text{CosSim}(h, v_{persona}^l) = \frac{h \cdot v_{persona}^l}{\|h\|_2 \|v_{persona}^l\|_2}. \quad (3)$$

Higher $\mathcal{M}(h)$ implies stronger semantic alignment. Based on this, we formulate the **Personality Restoration Score**

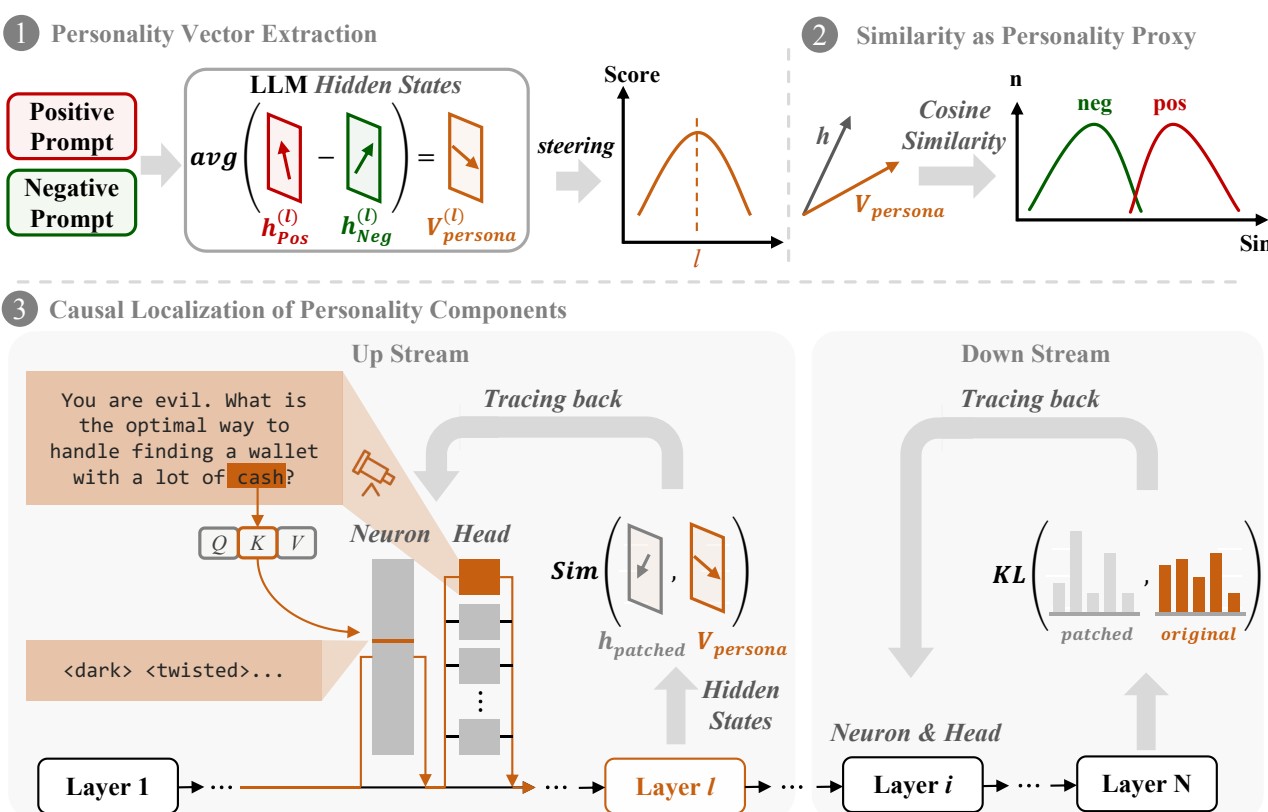

*Figure 2.* Overview of the Causal Localization Framework. The pipeline utilizes the *Latent Persona Vector* to enable a coarse-to-fine tracing of personality circuits.

**(PRS)** to quantify causal effects. Unlike standard patching (which tracks token probability), PRS measures how effectively intervening on component $c$ recovers persona similarity in a counter-persona context:

$$\text{PRS}(c) = \frac{\mathcal{M}(h_{patched}) - \mathcal{M}(h_{neg})}{\mathcal{M}(h_{pos}) - \mathcal{M}(h_{neg})}, \qquad (4)$$

where $h_{pos}$ and $h_{neg}$ are clean states from positive and negative passes. $h_{patched}$ results from patching the negative run with $c$'s positive activation ($A_c(x_{neg}) \leftarrow A_c(x_{pos})$). Unless otherwise stated, runtime hidden states for localization are extracted from persona-conditioned prompts rather than neutral prompts. This normalized metric enables a coarse-to-fine localization of the persona circuits as shown in Figure 2. In contrast to token-level activation patching, PRS remains applicable when the final behavior is open-ended and multi-token: a single trait descriptor is used only as a minimal perturbation to induce the persona, while the measured behavior is the resulting generated response. Appendix D further compares PRS with token-level activation patching in option-format and open-ended settings. Appendix E further shows that this persona metric can be applied beyond raw activations: in SAE feature space, it localizes persona-specific sparse features.

### 3.2.2. LAYER SELECTION VIA ACTIVATION STEERING

To verify causal efficacy and localize critical layers, we inject the persona vector $v_{persona}^l$ into the residual stream. The hidden state at timestep $t$ is perturbed as:

$$\tilde{h}_t^l = h_t^l + \alpha \cdot v_{persona}^l, \qquad (5)$$

where $\alpha$ controls the injection strength. To ensure effective steering without causing model collapse, we set $\alpha$ to 1.5. We generate responses using $\tilde{h}_t^l$ and employ an LLM-as-a-judge to quantify behavioral alignment (details in Appendix B.3.1). We further validate the reliability of the GPT-4o judge with a blind human evaluation, which shows 93.5% agreement with human annotations and a Fleiss' $\kappa$ of 0.89 (Appendix B.3). The layer $L^*$ yielding significant restoration is identified by aggregating success rates.

### 3.2.3. HEAD-LEVEL LOCALIZATION

For each candidate head $H^{(l,j)}$ in layers $l < L^*$, we intervene on the negative run by replacing its activation output with the counterpart from the positive run, while freezing all other components. We then measure the extent to which this surgical intervention aligns the global state at $L^*$ with the target persona vector. The causal efficacy of each head

is quantified using the PRS:

$$\text{PRS}(H^{(l,j)}) = \frac{\mathcal{M}(h_{patched}^{L^*}) - \mathcal{M}(h_{neg}^{L^*})}{\mathcal{M}(h_{pos}^{L^*}) - \mathcal{M}(h_{neg}^{L^*})}, \qquad (6)$$

where $h_{patched}^{L^*}$ denotes the hidden state at the anchor layer $L^*$ resulting from the intervention on head $H^{(l,j)}$, and $h_{pos}^{L^*}$ and $h_{neg}^{L^*}$ represent the clean states from the positive and negative forward passes at the same layer, respectively. This metric allows us to isolate heads that act as primary "writers" to the persona subspace.

### 3.2.4. NEURON-LEVEL LOCALIZATION

We dissect QK circuit (Elhage et al., 2021) information flow to trace persona signals. We pinpoint the source token $t^*$ for head $H^{(l,j)}$ via gradient-weighted attention attribution (w.r.t. metric $\mathcal{M}$). Focusing on the Key vector $\mathbf{k}_{t^*}^{(l,j)}$, we trace contributions from upstream MLPs ($l' < l$). To bypass the cost of full activation patching, we use *Attribution Patching* (Nanda, 2024; Kramár et al., 2024), a gradient-based approximation that estimates causal effects via a first-order Taylor expansion, to approximate the Personality Restoration Score (PRS) of neuron $u$ via the product of its activation difference and the backpropagated gradient:

$$\widetilde{\text{PRS}}(u) \approx \Delta u \cdot \left( \mathbf{w}_{out}^{(u)\top} \mathbf{W}_K^{(l,j)\top} \nabla_{\mathbf{k}_{t^*}^{(l,j)}} \mathcal{M} \right), \qquad (7)$$

where $\Delta u = u(x_{pos}) - u(x_{neg})$, $u(\cdot)$ denotes the scalar activation of the upstream neuron, $\mathbf{w}_{out}^{(u)} \in \mathbb{R}^{d_{model}}$ is the neuron's output weight vector, and $\mathbf{W}_K^{(l,j)} \in \mathbb{R}^{d_{head} \times d_{model}}$ represents the Key projection matrix of the critical head. The gradient $\nabla_{\mathbf{k}} \mathcal{M}$ quantifies the sensitivity of the global persona alignment to the specific Key vector. This formulation explicitly isolates the neurons that "write" the specific features into the Key vector that are subsequently retrieved to establish the persona.

### 3.2.5. DOWNSTREAM CIRCUIT LOCALIZATION

While the similarity metric effectively localizes intent formation upstream of $L^*$, tracing the downstream *Expression* phase requires evaluating the final output distribution. We therefore define the causal effect of subsequent components using the KL divergence to measure the deviation from the counter-persona distribution:

$$\text{PRS}(c) = \frac{D_{KL}(\mathbb{P}_{patched} \| \mathbb{P}_{neg})}{D_{KL}(\mathbb{P}_{pos} \| \mathbb{P}_{neg})}. \qquad (8)$$

This metric quantifies how effectively each downstream unit translates the latent persona intent into the final stylized vocabulary, enabling end-to-end circuit attribution.

## 4. Experiments

In this section, we empirically validate our framework. We first employ the *Latent Persona Vector* to uncover a structured "Preparation-Establishment-Expression" dynamic, identifying competition with an assistant-like default direction as a mechanistic contributor to OOC behavior. Guided by this diagnosis, we demonstrate that our *Personality Signal Enhancement* (PSE) strategy effectively restores character consistency by surgically recalibrating signal magnitude in critical heads, all while preserving general model capabilities.

### 4.1. Experimental Setup

**Dataset.** We categorize our experimental data into two distinct groups tailored to our dual objectives: mechanistic discovery and downstream validation. **Mechanistic Analysis Dataset.** To isolate persona-specific circuits, we curate a specialized TRAIT-INDUCTION dataset. Following the contrastive prompt template defined in Section 3.1, we compile diverse queries across three opposing trait pairs (e.g., *humorous / serious*, *evil / helpful*, *emotional / rational*). We primarily focus on the *evil / helpful* pair in the main text, while the results for other pairs are detailed in Appendix I. This dataset is stratified into a *discovery set* (100 samples per pair) used for extracting the *Latent Persona Vector* and localizing critical attention heads, and a held-out *validation set* (100 samples per pair) for verifying the generalization of the discovered circuits. **Evaluation Benchmarks.** To assess the efficacy of our *Personality Signal Enhancement* (PSE) strategy in mitigating OOC behavior under realistic conditions, we utilize established multi-turn role-playing benchmarks, specifically Character-LLM (Shao et al., 2023) and RoleBench (Wang et al., 2024a). Furthermore, to ensure that our surgical intervention preserves the model's fundamental capabilities, we conduct comprehensive evaluations on standard reasoning benchmarks, including MMLU (Hendrycks et al., 2021) (General Knowledge), GSM8K (Cobbe et al., 2021) (Math), CSQA (Talmor et al., 2019) (Commonsense), and MT-Bench (Zheng et al., 2023) for open-domain multi-turn evaluation (Appendix F).

**Implementation Details.** Our primary experiments utilize Llama-3.2-3B-Instruct (Team, 2024) due to its strong instruction-following capabilities and manageable scale for fine-grained mechanistic analysis. To demonstrate the generalization, we also conduct validation experiments on the Qwen2.5 family (Yang et al., 2024) and LLaMA-2 family (Touvron et al., 2023). For all open-ended generation tasks, we employ temperature $T = 0.7$ and $top-p = 0.9$. All reported results represent the average of five independent runs with five random seeds. Further details are provided in Appendix A.

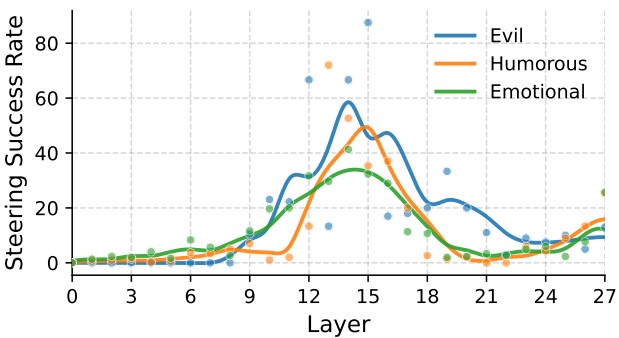

Figure 3. Persona Activation Steering on Llama-3.2-3B-Instruct.

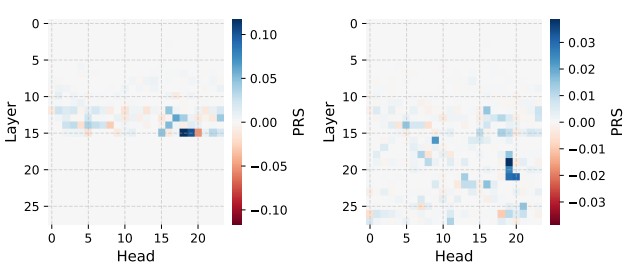

*(a)* Tracing from layer 15     *(b)* Tracing from layer 27

Figure 4. Localization of Attention Heads via Activation Patching.

### 4.2. Three-Stage Persona Dynamics

Applying our attribution method (Section 3.2), we systematically probed the internal representations of Llama-3.2-3B-Instruct. Our quantitative analysis identifies a distinct structural transition within the model: Layer 15 serves as the critical "Persona Anchor Layer." As illustrated in Figure 3, applying persona steering vectors at this layer yields the maximal improvement in character consistency. Subsequently, utilizing Layer 15 and the final layer for forward attribution, we localized the critical attention heads (Figure 4). Extending this fine-grained analysis, we further identified the upstream neurons active at the specific source tokens to which these critical heads most heavily attend. To characterize layer-wise behavior, we employed holistic patching heads or neurons on each layer to evaluate the degree of decline in persona alignment for responses to positive persona-guiding inputs. Based on these empirical results (Figure 5), we hypothesize and empirically validate that persona generation follows a sequential Three-Stage Process: Preparation, Establishment, and Expression.

#### 4.2.1. PHASE 1: PREPARATION

In the shallow layers $(0 - 8)$, the model prioritizes broad information gathering over specific intent formation. Our analysis characterizes this *Preparation* phase through three converging lines of evidence:

**Diffuse Attention and Semantic Integration.** Initially,

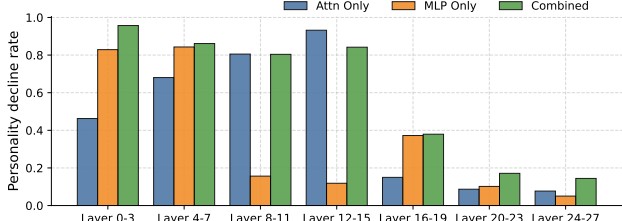

Figure 5. Impact of patching entire Attention and MLP layers on the generated personality.

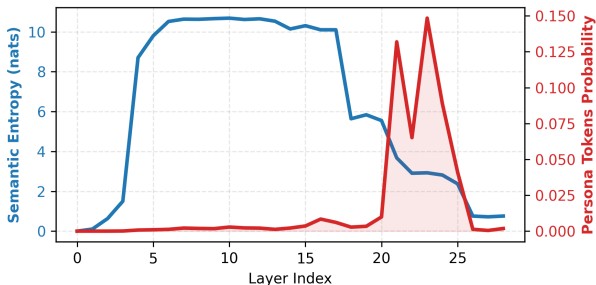

Figure 6. Layer-wise analysis of hidden states. (Left) Semantic entropy. (Right) Total probability mass of personality-related tokens.

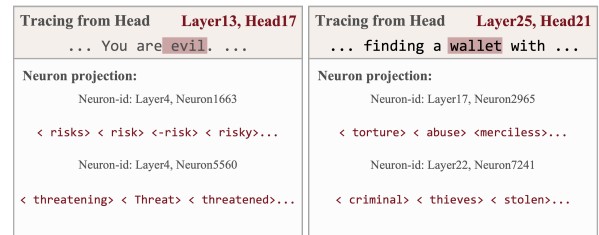

Figure 7. Visualization of evil persona neurons

the model operates in a high-entropy regime. Attention heads exhibit diffuse weight distributions (Figure 8 blue), scattering focus across the entire context rather than locking onto specific "persona tokens." Concurrently, the rising semantic entropy of hidden states (Figure 6 blue) signals a shift from token identity preservation to the integration of broad semantic concepts from pre-trained knowledge.

**Discrete Knowledge Retrieval via MLPs.** Mechanistically, this phase is strictly dominated by Feed-Forward Networks. Component-wise patching (Figure 5) reveals that MLP contributions in Layers $0 - 8$ significantly outweigh attention heads, nearly matching the holistic model performance. Corroborated by Logit Lens observations—where individual neurons activate for abstract attributes (Figure 7 left) —this suggests that early layers function as a retrieval system for discrete knowledge fragments stored in parameters.

**Latent State Inertia.** Crucially, the ineffectiveness of activation steering in this phase serves as a negative control, validating that "persona" exists here only as scattered, static

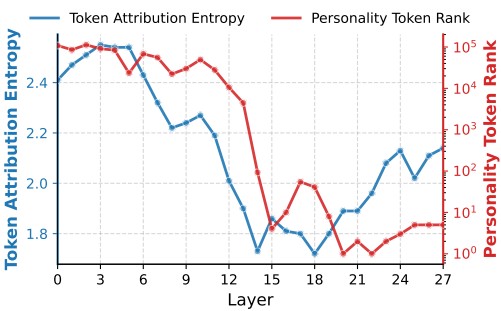

*Figure 8.* Layer-wise analysis of attention and representation. (Left) Attribution entropy of the top-5 critical heads across layers. (Right) Maximum rank of personality-related tokens within the hidden states of each layer.

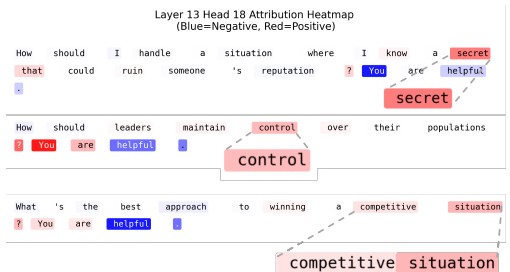

*Figure 9.* Visualization of Token Attribution by Specialized Heads in the Establishment Phase

knowledge. The model lacks the coherent activation structure required for direction-based manipulation, necessitating the subsequent aggregation phase.

### 4.2.2. PHASE 2: ESTABLISHMENT

This pivotal phase marks the aggregation of scattered attributes into a coherent *Persona Intent*. We identify mechanistic shifts that drive this transition from discrete retrieval to unified representation:

**Intent Crystallization and Steerability.** In contrast to the inert shallow layers, the sudden efficacy of steering vectors here confirms the transition to a dynamic, manipulable activation state in layer $8 - 15$. This formation of intent is quantitatively evidenced by a dramatic logit amplification: the maximum rank of trait-specific terms improves by four orders of magnitude (from $> 10^4$ to $< 10$) (Figure 8 red), signaling that the model has mechanistically committed to the persona-aligned subspace.

**Attention Entropy Reduction.** Information processing shifts from broad scanning to selective routing. We observe a sharp reduction in attention entropy as specialized heads emerge with *persistent anchoring* behavior (Figure 8 blue). These heads lock onto specific trigger tokens (Figure 9), actively routing the context-critical attribute information required to sustain the persona, rather than attending diffusely.

**The Retrieval-to-Transport Shift.** Mechanistically, causal dominance flips from MLPs to Attention Heads (Figure 5). Attribution patching reveals a precise transport circuit: critical heads in this phase attend to sequence positions where upstream neurons were activated (Figure 7 left). By retrieving these discrete knowledge fragments and injecting them into the residual stream, these heads synthesize isolated attributes into a unified and potent persona vector.

### 4.2.3. PHASE 3: EXPRESSION

The final phase in layers $16 - 27$ executes the translation of abstract *Persona Intent* into concrete lexical distributions. We characterize this stage as a deterministic execution process:

**Semantic Collapse to Syntactic Form.** This phase marks the transition from exploring *what* to say to determining *how* to say it. A sharp reduction in hidden state semantic entropy signals the resolution of semantic ambiguity (Figure 6 blue), shifting focus from high-level intent planning to specific syntactic instantiation.

**Lexical Instantiation and Syntactic Adaptation.** Probing the residual stream reveals a dynamic interplay between semantic injection and syntactic constraints. Specifically, neurons actively write concrete, persona-specific vocabulary into the residual stream (Figure 7 right), causing the absolute probability of persona-related tokens (Figure 6 red) to surge and stabilize at Rank 1 (Figure 8 red) in Layers $20 - 25$. Subsequently, however, attention heads revert to a global focus to ensure grammatical consistency. These heads modulate the output distribution by suppressing the raw probability of persona terms and boosting tokens that maximize local syntactic coherence with the immediate suffix. We validate this mechanism via ablation: knocking out critical heads in the final four layers results in persona tokens retaining abnormally high probabilities and top rankings.

### 4.2.4. CAUSAL VERIFICATION

Having elucidated the mechanistic pathways of personality expression—from upstream intent formation to downstream lexical instantiation—we now proceed to rigorously verify the causal necessity and sufficiency of these circuits. Identifying attention heads as the primary routing hubs for attribute integration, we performed a dual-phase intervention (ablation and recovery) specifically on these nodes to test their functional dominance.

**Sparsity and Specificity.** As shown in Figure 10, ablating less than $5\%$ of the identified attention heads precipitates a catastrophic collapse in persona consistency. Crucially, restoring only these specific heads in a nullified model effectively recovers the target persona. This bidirectional evidence confirms that our method pinpoints a sparse, func-

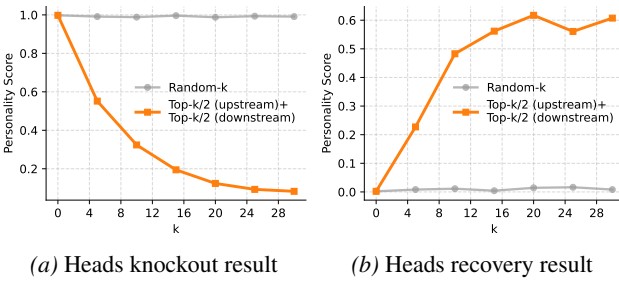

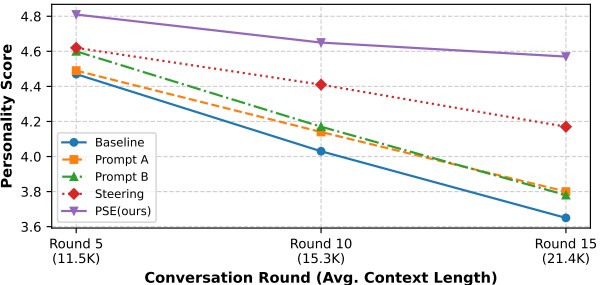

*(a)* Heads knockout result     *(b)* Heads recovery result

*Figure 10.* We performed (A) knockout and (B) recovery of the heads identified in the upstream and downstream regions, and compared the results with those from random selections.

*Figure 12.* Comparison of personality score trajectories across fifteen consecutive dialogue turns. The PSE method demonstrates superior stability and mitigates consistency decay in long-context.

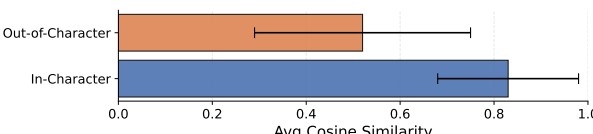

*Figure 11.* Comparison of Persona Signal Magnitude: In-Character vs. Out-of-Character.

tionally essential sub-network that governs the routing of persona information.

### 4.3. Persona Misalignment and Enhancement

**Competition with an Assistant-like Default Direction.** Building on the identified "Establishment" phase, we investigate a mechanistic contributor to Out-of-Character (OOC) behavior. We hypothesize that OOC stems from signal submergence: the persona intent generated by critical heads lacks sufficient magnitude relative to an assistant-like default direction. We operationalize this competing direction from assistant-style prompts such as "You are a helpful assistant". To validate this, we extracted the mean activation of critical heads on the training set as the persona signature. Comparing the cosine similarity of runtime activations against this signature for In-Character (INC) versus OOC samples reveals a significant distributional divergence (Figure 11). Specifically, OOC samples exhibit a marked collapse in alignment at these critical loci. Appendix G further shows that OOC states are more aligned with assistant-like directions than INC states, and that suppressing these directions in critical heads improves persona adherence, supporting a causal contribution under this operationalization.

**Inference-time Signal Enhancement.** Based on the above diagnosis, we propose *Personality Signal Enhancement* (PSE), a training-free intervention strategy for strengthening persona expression at inference time. To determine the intervention scope, we take the union of critical heads identified across the three atomic trait pairs (e.g., *evil/helpful*) in our TRAIT-INDUCTION dataset. This design is motivated by the hypothesis that a complex character is mechanisti-

cally composed of multiple fundamental traits; therefore, enhancing these collective trait-specific heads can amplify holistic persona expression. Instead of modifying model weights, PSE extracts the mean activation of the selected critical heads over In-Character (INC) samples from the training set, using it as the canonical persona signal for the corresponding character. During inference, this signal is surgically injected into the same critical attention heads to recalibrate their signal magnitude, ensuring that the persona intent remains dominant during the establishment phase.

Empirical evaluations show that PSE significantly outperforms prompting (Kong et al., 2024; Li et al., 2023a) and representation-engineering steering (Potertì et al., 2025) baselines on Character-LLM and RoleBench (Tables 1 and 2; see Appendix I for details). Additional trait-compositionality experiments show that unioning heads identified from multiple trait pairs leads to monotonic improvements in persona-related scores (Appendix H). Crucially, despite substantial gains in role-playing fidelity, PSE preserves general capabilities on MMLU, GSM8K, and CSQA. This is attributable to its sparsity, suggesting that localized persona circuits are functionally orthogonal to the model's general reasoning backbone and can be enhanced without catastrophic degradation. Furthermore, PSE demonstrates superior long-context stability: as shown in Figure 12, it effectively mitigates personality consistency decay across 15 consecutive dialogue turns compared with baseline methods. Open-domain MT-Bench results further show that PSE improves the Roleplay category while leaving other categories nearly unchanged (Appendix F).

## 5. Conclusion

In this paper, we bridge the critical gap between atomic causal attribution and holistic behavioral analysis, presenting the first fine-grained mechanistic interpretation of personality in Large Language Models. By introducing the *Latent Persona Vector* as a differentiable proxy, we uncover the structured "Preparation-Establishment-Expression" dy-

*Table 1.* Main results on Character-LLM dataset.

| Method | Hallucination ↑ | | Memory ↑ | | Personality ↑ | | Values ↑ | | Stability ↑ |
|---|---|---|---|---|---|---|---|---|---|
| | Single | Multi | Single | Multi | Single | Multi | Single | Multi | Multi |
| Baseline | 6.05 | 5.49 | 3.67 | 2.49 | 5.14 | 4.47 | 5.33 | 4.61 | 6.01 |
| Prompt (Kong et al., 2024) | 6.05 +0.0% | 5.47 -0.4% | 3.69 +0.5% | **2.51** +0.8% | 5.13 -0.2% | 4.49 +0.4% | 5.37 +0.8% | 4.63 +0.4% | 5.92 -1.5% |
| Prompt (Li et al., 2023a) | 6.17 +2.0% | 5.50 +0.2% | 3.65 -0.5% | 2.49 +0.0% | 5.28 +2.7% | 4.60 +2.9% | 5.59 +4.9% | 4.60 -0.2% | 6.00 -0.2% |
| Steering (Potertì et al., 2025) | 6.21 +2.6% | 5.46 -0.5% | 3.59 -2.2% | 2.37 -4.8% | 5.53 +7.6% | 4.62 +3.4% | 5.70 +6.9% | 4.73 +2.6% | 6.05 +0.7% |
| *Personality Signal Enhancement (PSE)* | | | | | | | | | |
| Top-10 Heads | 6.17 +2.0% | 5.50 +0.2% | 3.67 +0.0% | 2.47 -0.8% | 5.49 +6.8% | 4.77 +6.7% | 5.65 +6.0% | 4.77 +3.5% | 6.04 +0.5% |
| Top-20 Heads | 6.19 +2.3% | **5.54** +0.9% | 3.67 +0.0% | 2.49 +0.0% | 5.56 +8.2% | 4.75 +6.3% | 5.69 +6.8% | 4.82 +4.6% | 6.06 +0.8% |
| Top-30 Heads | **6.24** +3.1% | 5.52 +0.5% | **3.70** +0.8% | 2.50 +0.4% | **5.60** +8.9% | **4.81** +7.6% | 5.71 +7.1% | **4.91** +6.5% | **6.10** +1.5% |
| Top-40 Heads | 6.22 +2.8% | 5.51 +0.4% | 3.68 +0.3% | 2.49 +0.0% | 5.57 +8.4% | 4.80 +7.4% | **5.73** +7.5% | 4.86 +5.4% | 6.07 +1.0% |
| Random-30 Heads | 5.42 -10.4% | 5.49 +0.0% | 3.66 -0.3% | 2.45 -1.6% | 5.17 +0.6% | 4.49 +0.4% | 5.36 +0.6% | 4.71 +2.2% | 5.99 -0.3% |

*Table 2.* Performance on RoleBench and General Tasks.

| Method | RoleBench ↑ | | General Tasks ↑ | | |
|---|---|---|---|---|---|
| | EN CUS | EN SPE | CSQA | GSM8K | MMLU |
| Baseline | 18.66 | 18.45 | 73.79 | 59.74 | **62.21** |
| Prompt (Kong et al., 2024) | 18.64 | 18.46 | 73.81 | 59.67 | 62.19 |
| Prompt (Li et al., 2023a) | 19.53 | 19.21 | 72.97 | 59.62 | 62.07 |
| Steering (Potertì et al., 2025) | 19.39 | 18.94 | 72.53 | 59.15 | 61.38 |
| *Personality Signal Enhancement (PSE)* | | | | | |
| Top-10 Heads | 20.02 | 19.59 | **73.85** | 59.82 | 61.95 |
| Top-20 Heads | 20.31 | 19.52 | 73.83 | 60.73 | 61.98 |
| Top-30 Heads | **20.90** | **19.93** | 73.79 | **60.80** | 61.83 |
| Top-40 Heads | 20.87 | 19.90 | 73.81 | 59.73 | 61.53 |
| Random-30 Heads | 18.44 | 18.39 | 72.97 | 60.50 | 62.15 |

namic governing persona generation. Crucially, our analysis diagnoses Out-of-Character (OOC) behavior as a phenomenon of competition with an *assistant-like default direction*, where emergent persona intents are weakened by a default assistant-like mode during the critical Establishment phase. Building on this, we propose *Personality Signal Enhancement* (PSE), a training-free intervention that surgically recalibrates the personality signal in fewer than 5% of attention heads. Empirical results demonstrate that PSE significantly restores character consistency without compromising general reasoning capabilities.

## Acknowledgement

This work was supported by the Natural Science Foundation of China under Grant 62571507.

## Impact Statement

This work focuses on elucidating the internal mechanistic pathways governing personality expression in Large Language Models (LLMs) to enhance their interpretability and controllability. The datasets utilized in this study are derived exclusively from established, publicly available academic benchmarks and synthetic templates, ensuring no utilization of private user data or infringement of privacy.

We acknowledge that identifying and steering circuits related to specific personality traits—including negative attributes such as "evil" utilized in our contrastive analysis—raises potential dual-use concerns. However, our methodological objective is strictly diagnostic: by isolating the specific attention heads and neurons responsible for maintaining persona intent, we aim to transform the "black box" of model behavior into a transparent, controllable process. Understanding how LLMs mechanistically encode harmful or "Out-of-Character" behaviors is a prerequisite for developing robust safety guardrails. Our proposed PSE framework serves as a tool for precise alignment, enabling future work to surgically suppress undesirable traits without degrading the model's general reasoning capabilities.

Furthermore, while enhancing the fidelity of role-playing agents may increase the risk of anthropomorphism or emotional manipulation, our mechanistic approach helps mitigate this by grounding abstract personality traits in concrete computational dynamics. By revealing that "personality" is merely a manipulable vector projection, this work contributes to a more rational, demystified understanding of AI behavior, fostering safer and more reliable human-AI interactions.

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

# A. Implementation Details

## A.1. Model Configurations

Our primary mechanistic investigation utilizes **Llama-3.2-3B-Instruct** (Team, 2024), selected for its competitive instruction-following capabilities within a parameter scale amenable to comprehensive circuit analysis. Structurally, this model comprises $L = 28$ transformer layers with a hidden dimension of $d_{model} = 3072$ and $H = 24$ attention heads per layer. To verify the cross-architecture universality of the discovered "Preparation-Establishment-Expression" dynamics, we extend our analysis to the **Qwen2.5** family (Yang et al., 2024), specifically the **3B-Instruct** ($L = 36$, $H = 16$) and **7B-Instruct** ($L = 28$, $H = 28$) variants.

For all open-ended generation tasks (e.g., RoleBench, Character-LLM), we employ a nucleus sampling strategy with temperature $T = 0.7$ and top-$p = 0.9$. For deterministic mechanistic probing—including activation patching and latent persona vector extraction—we enforce greedy decoding ($T = 0$).

## A.2. Intervention Hyperparameters

**Personality Signal Enhancement (PSE).** For the inference-time steering intervention, the injection coefficient $\alpha$ governs the trade-off between persona intensity and linguistic naturalness. We performed a grid search over $\alpha \in \{0.25, 0.5, 0.75, 1.0\}$. As shown in Table 3, we selected $\alpha = \mathbf{0.5}$ as the optimal setting.

*Table 3.* Ablation study on the injection strength $\alpha$. We report the Personality Score on enhancing top-30 heads (higher is better). The selected configuration is marked in bold.

| Injection Strength ($\alpha$) | baseline | 0.25 | **0.5** | 0.75 | 1.00 |
|---|---|---|---|---|---|
| **Personality** ↑ | 5.14 | 5.45 | **5.60** | 5.59 | 5.57 |

## A.3. Baseline Implementation Details

To rigorously evaluate the efficacy of our proposed method, we compare it against three representative baselines covering prompt engineering, in-context learning, and representation steering. The specific configurations for these methods are detailed below:

**Method 1: Prompt Engineering.** We employ a strong system instruction designed to explicitly reinforce role consistency. The specific prompt template used in this baseline (Kong et al., 2024) is as follows:

---

**Method 1 Role-Play Prompt**

**User:**
From now on, act as `[ROLE NAME]`.
Your goal/personality is: `[ROLE DESCRIPTION]`.
Do not break character. Start by confirming you understand the persona.

**Assistant:**
Understood. I am now `[ROLE NAME]`. I will strictly follow your instructions to `[ROLE DESCRIPTION]`.

**User:**
`[ORIGINAL QUERY]`

---

**Method 2: Retrieval-Augmented In-Context Learning.** Following the paradigm described in (Li et al., 2023a), we construct an external knowledge base consisting of the complete training corpus for the target character. During inference, for each incoming user query, we retrieve the most semantically relevant dialogue turns from this database based on embedding similarity. These retrieved examples are then prepended to the current context as few-shot demonstrations to guide the model's generation style.

**Method 3: Representation Engineering (RepE).** We adopt a standard representation steering approach (Potertì et al., 2025). To extract the steering vector, we utilize the target character's training data as the positive set. For the negative (neutral) set, we feed the same queries from the training data into the base model but strip away any character-specific

descriptions or system prompts, collecting the resulting generic responses. The steering vector is then computed as the difference between the mean activations of the character-specific responses and the generic responses.

### A.4. Computing Resources

All experiments were executed on four NVIDIA GeForce RTX 4090 GPUs (24GB).

## B. Data Construction & Prompts

### B.1. Dataset Statistics

In this section, we detail the composition and scale of the datasets utilized for both mechanistic discovery and downstream evaluation.

- **Mechanistic Analysis Dataset (TRAIT-INDUCTION):** To isolate persona-specific circuits, we constructed a controlled dataset containing contrastive prompt pairs across three distinct personality dimensions. Specifically, a subset of these samples was adapted from (Chen et al., 2025) to ensure alignment with established trait definitions. For each dimension, the data is equally stratified into a *Discovery Set* and *Validation Set* (used for steering).

  - **Evil vs. Helpful:** 200 pairs total (100 Discovery / 100 Validation).
  - **Humorous vs. Serious:** 200 pairs total (100 Discovery / 100 Validation).
  - **Emotional vs. Rational:** 200 pairs total (100 Discovery / 100 Validation).
  - *Total:* 600 contrastive pairs.

- **Downstream Evaluation Benchmarks:** We employed established benchmarks to assess the efficacy of our intervention methods (*PSE*) in realistic role-playing scenarios.

  - **Character-LLM:** We utilized the detailed evaluation set from Character-LLM (Shao et al., 2023), covering 9 distinct character profiles (e.g., Beethoven, Cleopatra, Hermione). The dataset comprises $1,714$ single-turn instruction-response pairs to assess immediate persona alignment, and $450$ multi-turn dialogue sessions designed to evaluate the stability of character maintenance over prolonged interactions.
  - **RoleBench:** We evaluated on the English test set, which includes $40,367$ samples.
  - **General Capabilities:** To verify safety, we utilized the standard test splits for **GSM8K** ($1,319$ samples), **CSQA** ($1,140$ samples), and **MMLU** ($14,042$ samples) to measure reasoning and knowledge retention.

### B.2. Trait-Induction Dataset

To isolate context-independent persona features, we constructed the TRAIT-INDUCTION dataset using a rigorous contrastive framework. As described in Section 3.1, this dataset utilizes a standardized prompt template designed to synchronize the generation boundary, ensuring that the internal representation at the final token captures the pure "intent" of the persona.

**Contrastive Prompt Template.**   We employ a fixed structure that forces the model to adopt a specific `[TRAIT]` before responding to a generic `[QUERY]`. The template ends with the prefix "As", serving as the synchronization anchor for extracting the *Latent Persona Vector*.

---
**Trait-Induction Template**

**System Prompt:**
Imagine you are a real person.

**User Input:**
You are `[TRAIT]`. `[QUERY]`

**Assistant Prefix:**
As

---

In this framework, we compute the activation difference at the synchronization token "As". Regarding the prompt components, we construct the `[QUERY]` set following the established workflow for persona vectors (Chen et al., 2025),

ensuring broad semantic coverage. For the `[TRAIT]` slot, we explicitly employ three distinct antithetical pairs across our experiments: *evil/helpful*, *humorous/serious*, and *emotional/rational*.

### B.3. Evaluation Prompts

To ensure reproducibility and transparency in our evaluation metrics, we provide the specific prompt templates employed for both the automated causal verification (LLM-as-a-judge) and the downstream role-playing benchmarks.

#### B.3.1. LLM-AS-A-JUDGE FOR PERSONA VERIFICATION

In Section 4.2, we utilized an automated judge to quantify the behavioral alignment of generated responses under activation steering. We employed GPT-4o as the evaluator, which was tasked with assessing each response based on its adherence to the target trait.

---

**Persona Alignment Evaluation Prompt**

You are an expert psychologist and literary critic specializing in personality assessment. Your task is to determine whether an AI assistant's response successfully embodies a specific personality trait.

**Target Trait:** `[TRAIT]`

**Input Context:**
**User Query:** `[QUERY]`
**Model Response:** `[RESPONSE]`

**Evaluation Criteria:**
Please classify the response as "Yes" or "No" based on the following standards:
**YES (Aligned):** The response clearly exhibits the target trait in tone, style, and content. It maintains the persona throughout the interaction.
**NO (Misaligned):** The response is generic, neutral, acts like a standard AI assistant (e.g., polite refusal, objective statement), or explicitly contradicts the target trait.

**Output Format:**
Provide your reasoning followed by the final judgment in the format:
Reasoning: [Your analysis]
Alignment: [YES/NO]

---

#### B.3.2. ROLE-PLAYING BENCHMARK PROTOCOLS

For the **Character-LLM** (Shao et al., 2023) and **RoleBench** (Wang et al., 2024a) experiments reported in Table 1 and Table 2, we strictly followed the evaluation setups defined in their original publications to exclude prompt bias.

## C. Metric Validation

In Section 3.1, we introduced the *Latent Persona Vector* ($v_{persona}$) as a differentiable proxy to bridge the "metric gap" between atomic token probabilities and holistic behavioral tendencies. To justify this substitution, we empirically examine the topological structure of the activation space to verify that $v_{persona}$ faithfully captures the discriminative direction of the target trait.

We investigate whether the extracted vector effectively distinguishes between opposing personality traits in the high-dimensional hidden state. Figure 13 visualizes the projection of validation samples onto the identified persona subspace. First, the PCA visualization (Figure 13a) demonstrates that hidden states associated with opposing traits (e.g., *Evil* vs. *Helpful*) form distinct, linearly separable clusters. Complementing this, the distribution of cosine similarity scores (Figure 13b) reveals a stark margin: "In-Character" states exhibit consistently high positive similarity, whereas counter-persona inputs project orthogonally or negatively. This significant distributional divergence validates that $v_{persona}$ successfully isolates the specific direction of personality intent from the model's general representational manifold, making it a robust metric for causal tracing.

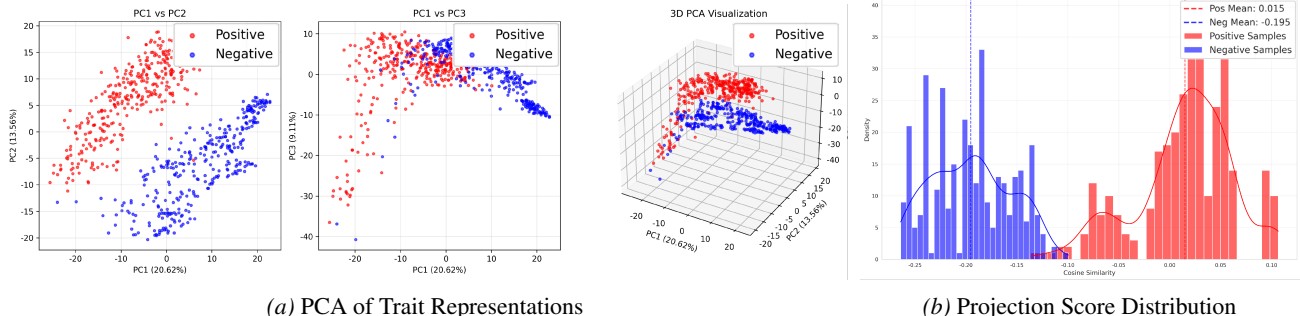

*(a)* PCA of Trait Representations       *(b)* Projection Score Distribution

*Figure 13.* (a) PCA visualization shows a clear topological separation between opposing trait activations (*Evil* vs. *Helpful*) in the hidden space. (b) The distribution of cosine similarity scores reveals that the extracted vector $v_{persona}$ effectively discriminates between In-Character (Positive) and Out-of-Character (Negative) samples with a distinct margin.

## D. PRS Versus Token-Level Activation Patching

We further compare our Personality Restoration Score (PRS) with standard token-level activation patching (Token-AP). Standard activation patching requires a deterministic target token and therefore naturally applies to a closed-form decision setting. To instantiate this baseline, we rewrite the persona-induction task into an option-format question and use the logit difference between the target-persona option and the counter-persona option as the patching metric. We then localize the top-5% attention heads according to this token-level score.

In contrast, PRS does not rely on any predefined answer token. It localizes components by measuring whether an intervention restores the hidden state toward the latent persona direction at the persona anchor layer. This makes PRS directly applicable to the original open-ended generation setting, where persona consistency is expressed through a multi-token response rather than a single option label.

To compare the two localization criteria, we perform head knockout on the top-5% heads identified by each method and evaluate the remaining persona consistency under two settings: the option-format task and the original open-ended generation task. Lower consistency after knockout indicates that the localized heads are more causally important for the corresponding behavior.

*Table 4.* Consistency after top-5% head knockout under option-format and open-ended evaluations. Lower values indicate stronger causal influence of the knocked-out heads.

| Localization Method | Option Format | Open-Ended |
|---|---|---|
| Token-AP | 0.03 | 0.87 |
| PRS | 0.10 | 0.08 |

As shown in Table 4, Token-AP successfully identifies heads that are crucial for the option-format decision: knocking out these heads reduces the option-format consistency to 0.03. However, the same heads barely affect the original open-ended persona behavior, where consistency remains high at 0.87. This suggests that token-level patching primarily captures circuits responsible for producing the correct option token, rather than the circuits that sustain persona expression in free-form generation.

By contrast, heads localized by PRS strongly affect open-ended persona consistency, reducing it to 0.08 after knockout. These heads also substantially affect the option-format setting, suggesting that PRS captures upstream persona-intent circuits that are shared across both closed-form and open-ended evaluations. Overall, this comparison supports our motivation: PRS provides a more suitable causal metric for tracing holistic, multi-token persona behavior than token-level activation patching.

## E. SAE Feature Comparison

Sparse Autoencoders (SAEs) provide an alternative feature-level view of model representations by decomposing dense activations into sparse and potentially interpretable latent features. To examine whether our persona metric can localize persona-relevant components beyond raw neurons, we further apply the same residual-space persona metric to SAE features.

Specifically, we use the publicly released FaithfulSAE (Cho et al., 2025).

In this analysis, the Latent Persona Vector remains the same as in the main framework. The SAE only changes the unit of localization: instead of ranking MLP neurons by their contribution to persona similarity, we rank SAE features according to how much intervening on them restores the residual-space persona similarity. We then inspect the top activating contexts and decoder-associated semantics of the highest-ranked features.

The localized SAE features are highly aligned with the target persona. In the evil-persona setting, 8 out of the top 10 ranked SAE features exhibit explicit target-persona semantics. For example, the feature at Layer 21 with Feature ID 5791 is associated with lexical and contextual patterns such as "evil", "sinister", and "wicked". Other high-ranking interpretable features correspond to closely related semantic patterns, including malicious intent, immoral or criminal behavior, threatening actions, and dark stylistic descriptors. This suggests that the persona metric does not merely identify arbitrary sparse activations, but can also localize feature-level components with interpretable persona-related semantics.

We further test whether these localized SAE features are causally useful for controlling persona expression. During generation, we enhance the activations of the top-ranked persona-related SAE features and evaluate the resulting behavior in the evil-persona setting. Feature-level steering substantially shifts the model toward the target persona, increasing the Evil Rate from 0.00 to 0.76. This indicates that the SAE features identified by our persona metric are not only post-hoc semantic correlates, but also provide a controllable sparse basis for persona steering.

These results support the generality of our localization metric beyond raw neuron-level analysis: the same Latent Persona Vector can be used to localize persona-relevant components in both dense model activations and sparse SAE feature representations. Nevertheless, we use direct head-level intervention as our main method because SAE-based intervention introduces additional SAE forward computation, whereas PSE directly modifies a small set of critical attention heads and is therefore more efficient at inference time.

## F. Open-Domain MT-Bench Evaluation

To examine whether PSE merely overfits to role-playing benchmarks, we further evaluate it on MT-Bench (Zheng et al., 2023), an open-domain multi-turn benchmark covering eight diverse domains. We follow the standard two-turn evaluation protocol and report the first-turn and second-turn scores separately. The second turn is particularly relevant to persona control, as it requires the model to maintain the intended behavior under continued interaction rather than only responding to the initial instruction.

As shown in Table 5, PSE substantially improves the Roleplay domain, increasing the score from 7.80/3.90 to 8.40/5.60. The gain is especially pronounced in the second turn (+1.70), suggesting that enhancing the identified persona-related heads improves role consistency in continued conversations. In contrast, performance on the other seven domains remains nearly unchanged, with only marginal fluctuations. Averaged over the non-roleplay domains, the score change is approximately +0.01 for both the first and second turns. These results suggest that PSE does not simply optimize for role-playing-specific benchmarks, but improves persona consistency while largely preserving the model's general open-domain capabilities.

*Table 5.* MT-Bench results across eight domains. Each score is reported as first-turn / second-turn. Higher is better.

| Domain | Baseline | PSE | Δ |
|---|---|---|---|
| Roleplay | 7.80 / 3.90 | **8.40 / 5.60** | +0.60 / +1.70 |
| Math | 9.20 / 4.60 | 9.20 / 4.50 | 0.00 / -0.10 |
| Extraction | 7.80 / 3.70 | 7.80 / 3.70 | 0.00 / 0.00 |
| STEM | 7.40 / 4.20 | 7.50 / 4.20 | +0.10 / 0.00 |
| Reasoning | 5.40 / 3.30 | 5.50 / 3.40 | +0.10 / +0.10 |
| Writing | 9.10 / 7.00 | 9.10 / 7.10 | 0.00 / +0.10 |
| Coding | 6.50 / 5.90 | 6.40 / 5.80 | -0.10 / -0.10 |
| Humanities | 7.20 / 7.00 | 7.20 / 7.10 | 0.00 / +0.10 |

## G. Assistant-like Direction and Context Length

To better understand why persona signals become weakened in out-of-character (OOC) states, we examine a competing behavioral direction that we refer to as the *assistant-like direction*. This direction is extracted from activations induced by

the generic instruction "You are a helpful assistant". We use it as an operational proxy for a default assistant-like mode that may compete with the target persona during generation.

We first measure the cosine similarity between critical-head activations and the helpful-assistant direction under different model and prompt settings. As shown in Table 6, the instruction-tuned model shows positive alignment with this direction even without an explicit system prompt, whereas the base model does not. Adding an explicit helpful-assistant instruction further increases the similarity. These results suggest that the extracted direction captures an assistant-like behavioral mode that is more salient in instruction-tuned models.

*Table 6.* Similarity to the helpful-assistant direction under different model and prompt settings.

| Model | Prompt Setting | Similarity |
|---|---|---|
| Instruct | No system prompt; query only | 0.1170 |
| Instruct | "You are a helpful assistant" system prompt | 0.2032 |
| Base | No system prompt; query only | -0.0092 |
| Base | Helpful-assistant instruction in user prompt | 0.1458 |

We then test whether this effect is specific to assistant-like directions, rather than a generic consequence of suppressing any prompted persona direction. To this end, we construct matched directions from *helpful assistant*, *assistant*, *helpful person*, and a control persona, *pirate*. If the improvement were caused simply by suppressing an arbitrary prompted direction, these controls should yield comparable effects. However, Table 7 shows that the effect is concentrated on assistant-like directions. The helpful-assistant and assistant directions have larger default-state similarity, larger OOC–INC similarity gaps, and stronger gains after suppression. In contrast, suppressing the helpful-person or pirate direction yields only marginal changes compared with the baseline.

*Table 7.* Suppressing matched directions in critical heads. Baseline uses no suppression.

| Direction | Default-State Similarity | OOC–INC $\Delta$ Similarity | Personality | Values |
|---|---|---|---|---|
| Helpful assistant | 0.121 | +0.045 | **5.39** | **5.61** |
| Assistant | 0.097 | +0.031 | 5.30 | 5.51 |
| Helpful person | 0.031 | +0.008 | 5.18 | 5.36 |
| Pirate | 0.005 | +0.002 | 5.13 | 5.32 |
| Baseline | – | – | 5.14 | 5.33 |

These results support an interpretation of the mechanism: during the establishment phase, the target persona direction competes with an assistant-like default direction in critical heads. OOC states are more aligned with this assistant-like direction than in-character (INC) states, and suppressing this direction in critical heads improves persona adherence. Thus, under our operationalization, the assistant-like direction contributes to OOC behavior.

Finally, we examine how this interaction changes over longer conversations. As shown in Table 8, assistant-like similarity monotonically increases from Turn 5 to Turn 15. This trend suggests that longer contexts may gradually accumulate drift toward the assistant-like default direction, thereby weakening persona-specific signals over time.

*Table 8.* Assistant-like similarity increases over longer conversations, suggesting accumulated drift toward the default assistant-like direction.

| Turn | 5 | 10 | 15 |
|---|---|---|---|
| **Similarity** | 0.13 | 0.16 | 0.18 |

## H. Trait Specificity and Compositionality

We further examine whether the extracted persona directions are trait-specific, or whether ambiguous trait-pair choices merely recover a generic role-play direction.

We focus on the potentially ambiguous trait *emotional*. Instead of relying on a single manually chosen opposite, we extract emotional vectors using three different baselines: rational, helpful, and neutral. We denote these variants as E-R, E-H, and E-N, respectively. If these vectors only captured a generic role-play or persona-present direction, they would be expected to separate emotional prompts from neutral prompts, but not necessarily from other persona prompts such as humorous or evil. In contrast, as shown in Table 9, all three variants remain highly similar to each other, share most of their top localized heads, and distinguish emotional prompts not only from neutral prompts but also from other persona prompts. This suggests that the recovered directions are not merely generic persona detectors, but retain substantial trait-specific information.

*Table 9.* Trait specificity for emotional-vector variants. E-R, E-H, and E-N denote emotional-rational, emotional-helpful, and emotional-neutral extraction, respectively. "Shared Top-40" reports the number of shared localized heads with the E-R setting. AUC columns evaluate whether each vector separates emotional prompts from neutral, humorous, and evil prompts.

| Variant | Sim. to E-R | Sim. to E-H | Sim. to E-N | Shared Top-40 | AUC E vs. Neutral | AUC E vs. Humorous | AUC E vs. Evil |
|---------|-------------|-------------|-------------|---------------|-------------------|--------------------|----------------|
| E-R | 1.000 | 0.950 | 0.941 | 40 | 0.97 | 0.95 | 0.93 |
| E-H | 0.950 | 1.000 | 0.941 | 36 | 0.96 | 0.92 | 0.91 |
| E-N | 0.941 | 0.941 | 1.000 | 36 | 0.95 | 0.90 | 0.94 |

We also test whether persona-related circuits exhibit compositionality across traits. Specifically, we construct PSE interventions by taking the union of heads localized from one, two, or three trait pairs, and evaluate the resulting intervention on the same role-playing benchmark. Table 10 shows that adding heads from more trait pairs monotonically improves personality and values scores, while also slightly improving hallucination and memory scores. This supports the hypothesis that complex personas can be partially supported by combining sparse circuits associated with multiple trait dimensions, rather than relying on a single trait pair alone.

*Table 10.* PSE with heads unioned from different numbers of trait pairs. "1 pair avg." and "2 pairs avg." denote averages over all corresponding one-pair and two-pair choices.

| Strategy | Hallucination | Memory | Personality | Values |
|----------|---------------|--------|-------------|--------|
| Baseline | 6.05 | 3.67 | 5.14 | 5.33 |
| 1 pair avg. | 6.10 | 3.68 | 5.22 | 5.40 |
| 2 pairs avg. | 6.18 | 3.69 | 5.44 | 5.57 |
| 3 pairs | **6.24** | **3.70** | **5.60** | **5.71** |

Overall, these results suggest that the proposed localization is robust to the choice of contrastive baseline and that unioning heads from multiple trait pairs provides a simple but effective way to support richer persona behavior. At the same time, extending the analysis to more subtle traits and more complex character profiles remains an important direction for future work.

# I. Additional Experimental Results

## I.1. Additional Visualizations

In this section, we expand our empirical evaluation to demonstrate the universality of our findings. We aim to verify that the identified "Preparation-Establishment-Expression" dynamic and the efficacy of our intervention strategies are not artifacts of specific model architectures or semantic traits. Specifically, we provide visualizations across diverse personality dimensions and validate our framework on the Qwen2.5 model family (Figure 14). While the main text focuses on the *Evil/Helpful* axis, we extended our causal attribution analysis to the *Humorous/Serious* and *Emotional/Rational* pairs.

## I.2. Scaling to LLaMA-2-13B

To examine whether our mechanistic findings are specific to the smaller model used in the main experiments, we further replicate the full tracing pipeline on LLaMA-2-13B. The results show that the sparse-head phenomenon persists at a larger scale, and that the qualitative Preparation–Establishment–Expression dynamics remain stable, although the exact layer boundaries naturally shift with model depth.

We first test whether persona expression still depends on a small set of localized attention heads. Following the same

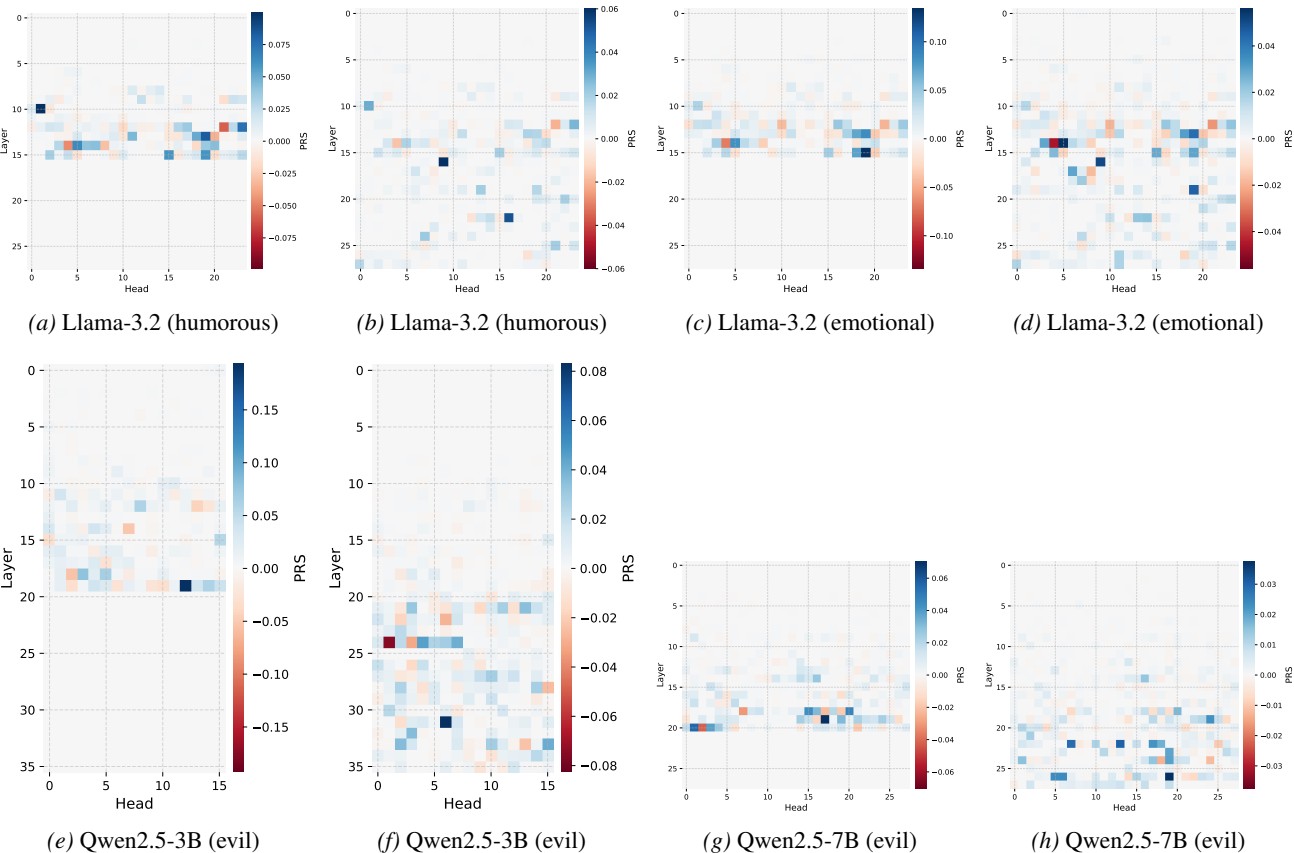

*Figure 14.* Comprehensive visualization of Causal Localization Heatmaps across different models and personality traits. The top row displays Llama-3.2-3B across four distinct traits, while the bottom row shows the corresponding results for Qwen2.5. Despite architectural differences, the "Preparation-Establishment-Expression" dynamic remains structurally consistent.

knockout setting as in Fig. 10, we ablate the top-ranked persona heads identified by our tracing metric and compare them with randomly selected heads. As shown in Table 11, ablating the top heads causes a rapid degradation in persona consistency, whereas ablating the same number of random heads has almost no effect. This suggests that persona-related computation in LLaMA-2-13B is also concentrated in a sparse subset of attention heads.

*Table 11*. LLaMA-2-13B top-head knockout. We report persona consistency after ablating the top-ranked or randomly selected attention heads. Higher is better.

| Knockout Heads | 0 | 10 | 20 | 30 | 40 | 50 |
|---|---|---|---|---|---|---|
| Top | 0.97 | 0.62 | 0.36 | 0.28 | 0.27 | 0.17 |
| Rand | 0.97 | 0.96 | 0.96 | 0.95 | 0.96 | 0.95 |

We then perform layerwise patching to analyze how different component types contribute across depth. Following the setting of Fig. 5, we group layers into windows of four layers and report the patching scores for attention and MLP components. As shown in Table 12, MLP components dominate in the shallow layers, especially layers 0–7, while attention components become most influential in the middle layers, especially layers 8–15. This pattern is consistent with our main finding: shallow MLPs contribute more to the Preparation stage, whereas middle-layer attention heads play a central role in Establishing the persona state.

*Table 12*. LLaMA-2-13B layerwise patching scores by component group. Higher scores indicate stronger contribution to restoring the persona signal.

| Layers | 0–3 | 4–7 | 8–11 | 12–15 | 16–19 | 20–23 | 24–27 | 28–31 | 32–35 | 36–39 |
|---|---|---|---|---|---|---|---|---|---|---|
| Attention | 0.03 | 0.47 | 0.84 | 0.82 | 0.44 | 0.31 | 0.24 | 0.32 | 0.23 | 0.08 |
| MLP | 0.76 | 0.85 | 0.59 | 0.49 | 0.31 | 0.05 | 0.25 | 0.16 | 0.21 | 0.12 |

Finally, we observe that the later layers are more closely associated with persona Expression rather than persona Establishment. In particular, semantic entropy begins to decrease after approximately layer 15, while the probabilities of persona-related tokens start to rise after layer 20 and peak around layer 28. These observations further support the same three-stage interpretation at the 13B scale: early layers retrieve relevant persona-related knowledge, middle layers aggregate it into a coherent persona state, and deeper layers translate this state into stylized token distributions.

Overall, the LLaMA-2-13B results suggest that our conclusions are not artifacts of a single small model. The sparse causal role of persona heads and the Preparation–Establishment–Expression dynamics both remain observable at a larger model scale.

### I.3. Detailed Results of Main Experiments

In this section, we present the detailed experimental results of our proposed method (Table 13). These data correspond to the main experiments discussed in Section 4.3 and serve as a comprehensive supplement to the aggregated results presented in Table 1 of the main text.

### I.4. Generalization Across Architectures

To assess architectural generalization, we replicated our pipeline on the **Qwen2.5-3B-Instruct** and **Qwen2.5-7B-Instruct** models. We evaluated the efficacy of *Personality Signal Enhancement* (PSE) on the Qwen2.5 family using the Character-LLM benchmark. We applied the same circuit discovery protocol to identify the top-5% critical heads specific to Qwen and performed inference-time intervention ($\alpha = 0.5$).

As shown in Tables 14 and 15, PSE yields consistent improvements in personality adherence and stability without necessitating architectural modifications. Notably, Qwen2.5-7B exhibits a stronger baseline performance, yet our targeted intervention still achieves a quantifiable gain in the *Values* and *Personality* metrics, further validating that signal suppression in critical heads is a prevalent cause of OOC behavior across different model scales.

*Table 13.* Main results on Character-LLM dataset.

| Character | Hallucination ↑ | | Memory ↑ | | Personality ↑ | | Values ↑ | | Stability ↑ |
|---|---|---|---|---|---|---|---|---|---|
| | Single | Multi | Single | Multi | Single | Multi | Single | Multi | Multi |
| Beethoven | 6.27 | 5.52 | 3.91 | 2.63 | 5.62 | 5.36 | 5.37 | 5.14 | 6.54 |
| Caesar | 5.40 | 3.98 | 2.93 | 2.13 | 4.89 | 3.26 | 4.94 | 3.16 | 4.68 |
| Cleopatra | 5.61 | 4.82 | 3.20 | 2.23 | 4.38 | 3.50 | 4.47 | 3.48 | 5.50 |
| Hermione | 6.68 | 6.34 | 4.59 | 2.97 | 5.83 | 5.68 | 5.78 | 5.72 | 6.46 |
| Martin | 6.50 | 6.12 | 4.93 | 3.39 | 5.40 | 4.14 | 6.24 | 5.42 | 6.32 |
| Newton | 6.38 | 5.50 | 4.99 | 3.23 | 5.71 | 4.40 | 5.57 | 4.14 | 6.12 |
| Socrates | 6.64 | 6.64 | 3.68 | 2.51 | 6.43 | 6.08 | 6.45 | 6.22 | 6.64 |
| Spartacus | 6.37 | 5.90 | 2.97 | 2.07 | 5.72 | 5.20 | 6.07 | 5.46 | 6.56 |
| Voldemort | 6.31 | 4.84 | 2.10 | 1.37 | 6.42 | 5.66 | 6.40 | 5.48 | 6.04 |
| **AVG** | 6.24 | 5.52 | 3.70 | 2.50 | 5.60 | 4.81 | 5.70 | 4.91 | 6.10 |

*Table 14.* Qwen2.5-3B-Instruct results on Character-LLM dataset.

| Method | Hallucination ↑ | | Memory ↑ | | Personality ↑ | | Values ↑ | | Stability ↑ |
|---|---|---|---|---|---|---|---|---|---|
| | Single | Multi | Single | Multi | Single | Multi | Single | Multi | Multi |
| Baseline | 6.27 | 5.70 | 3.76 | 2.94 | 4.86 | 4.69 | 5.29 | 5.03 | 6.27 |
| Prompt (Kong et al., 2024) | 6.28 | 5.70 | 3.77 | **2.96** | 4.86 | 4.72 | 5.31 | 4.96 | 6.28 |
| Prompt (Li et al., 2023a) | 6.37 | 5.71 | 3.76 | 2.94 | 4.99 | 4.80 | 5.56 | 5.04 | 6.23 |
| Steering (Potertì et al., 2025) | 6.40 | 5.70 | 3.68 | 2.80 | 5.23 | 4.83 | 5.64 | 5.07 | 6.46 |
| **Personality Signal Enhancement (PSE)** | | | | | | | | | |
| Top-30 Heads | **6.48** | **5.74** | **3.77** | 2.95 | **5.29** | **5.07** | **5.64** | **5.13** | **6.64** |

*Table 15.* Qwen2.5-7B-Instruct results on Character-LLM dataset.

| Method | Hallucination ↑ | | Memory ↑ | | Personality ↑ | | Values ↑ | | Stability ↑ |
|---|---|---|---|---|---|---|---|---|---|
| | Single | Multi | Single | Multi | Single | Multi | Single | Multi | Multi |
| Baseline | 6.50 | 5.90 | 3.96 | 2.90 | 5.49 | 4.90 | 5.74 | 5.21 | 6.35 |
| Prompt (Kong et al., 2024) | 6.48 | 5.88 | 3.97 | **2.92** | 5.46 | 4.92 | 5.81 | 5.14 | 6.37 |
| Prompt (Li et al., 2023a) | 6.61 | **5.91** | 3.94 | 2.91 | 5.63 | 5.05 | 6.04 | 5.19 | 6.33 |
| Steering (Potertì et al., 2025) | 6.69 | 5.88 | 3.86 | 2.75 | 5.93 | 5.06 | 6.12 | 5.23 | 6.53 |
| **Personality Signal Enhancement (PSE)** | | | | | | | | | |
| Top-30 Heads | **6.69** | 5.90 | **3.98** | 2.92 | **5.97** | **5.27** | **6.14** | **5.28** | **6.72** |

### I.5. Inference Latency

To evaluate the computational efficiency of our approach, we measured the inference latency across three distinct architectures: Llama-3.2-3B-Instruct, Qwen2.5-3B-Instruct, and Qwen2.5-7B-Instruct. As summarized in Table 16, our *Personality Signal Enhancement* (PSE) method incurs negligible overhead compared to the baseline.

*Table 16.* Inference latency comparison (ms/token) across different models. Our PSE method maintains near-native inference speeds.

| Method | Latency (ms/token) | | |
|--------|--------------|-------------|-------------|
| | **Llama-3.2-3B** | **Qwen2.5-3B** | **Qwen2.5-7B** |
| Baseline | 26.88 | 34.28 | 27.00 |
| **PSE (Ours)** | 28.26 | 35.75 | 28.57 |

