# OpenReview forum: "Tracing the Persona Circuit: How Large Language Models Encode and Express Character Traits"
_ICML.cc/2026/Conference — ICML 2026 regular_

### Official Review · Reviewer_veB1 · 2026-02-19

**Soundness:** 3
**Presentation:** 2
**Significance:** 2
**Originality:** 2
**Overall Recommendation:** 3
**Confidence:** 4

**Summary:**

The paper proposed Latent Persona Vector to help analyze the multi-token behavioral tendency in persona settings. By analyzing layer-wise behaviors, they reveal a structured "Preparation-Establishment-Expression" dynamic. Then, they propose a method to make the persona more consistent without training.

**Compliance With Llm Reviewing Policy:**

Affirmed.

**Final Justification:**

I appreciate the rebuttal. However, as they suggested, some of the statements were overclaimed, which I am still a bit concerned about. Therefore, I chose to keep my score.

**Key Questions For Authors:**

1.  In Lines 114–118, projection is mentioned. Projection onto which direction is performed? Is the residual stream explicitly projected onto the latent persona vector ($v_{persona}$)?

2. The paper argues that persona cannot be measured at the level of atomic tokens. However, it seems that the trait-induction prompt still uses single-token descriptors (e.g., “evil”). Could you elaborate on this?

3. Could the authors elaborate on why PRS enables a “coarse-to-fine localization”, whereas standard activation patching cannot? What specific property of the metric makes the localization finer?

4. When computing the runtime hidden state (h), is this extracted under a persona-conditioned prompt or a neutral prompt? Particularly, I am not very sure about Figure 2.2

5. How are “positive” and “negative” traits defined? For example, in pairs like “emotional/rational,” the semantic polarity seems weak. Could human bias in trait pairing affect the extracted persona direction? Could you provide more examples about the traits?

6. How is “consistency” evaluated in Section 4.2? When Layer 15 is identified as the “persona anchor layer,” does consistency refer to cosine similarity improvement, KL divergence, or an external judge score?

7. In Lines 370–373, the paper attributes OOC behavior to intrinsic assistant priors. How do the authors operationalize or measure these priors? For example, is “helpful assistant” treated as an implicit persona baseline in the system prompt?

**Limitations:**

yes

**Strengths And Weaknesses:**

## Strength
1. The layer-wise analysis provides a clear illustration of prior empirical findings on layer functions in interpretability work.
2. The paper proposes an interesting structured Preparation – Establishment – Expression framework.
## Weakness
1. The presentation could be improved, especially in Section 3. In particular, it would be helpful to clarify which exact prompts are used when extracting hidden states, how “consistency” is computed, and what direction the residual stream is being projected onto. Providing more concrete examples of positive/negative classification of the persona would also be helpful. Please see the detailed questions below.
2. The use of cosine similarity as a metric for persona alignment is intuitive and may capture sequence-level persona. However, it may also reflect unrelated or noisy factors, such as prompt formatting, task similarity, or dataset-specific artifacts. Also, the method may be sensitive to poorly aligned pairs. The paper would benefit from discussing these limitations and giving more examples about the pairs.
3. Section 4.3 ("Signal Submergence in Critical Heads") would benefit from more discussion. it is claimed that persona signals fail to override intrinsic assistant priors, but it is unclear why lower cosine similarity directly implies an inability to override. Lower cosine similarity is more likely to show that it is not aligned with the character, but it seems less connected with the assistant priors. It would be better with more justification.
4. The proposed method seems to be limited to getting the latent persona vector for pairs. It extracts a latent persona vector from positive/negative trait pairs, effectively representing a direction between opposites rather than a single standalone persona. This raises concerns for traits without clear antonyms (e.g., "curious") and for cases where persona opposites are not bijective. It remains unclear how the method generalizes when multiple opposites exist in different contexts.

---

> ### Author Rebuttal · Authors · 2026-03-31
>
> We thank the reviewer veB1 and respond point-by-point.
>
> > **W1, Q1, Q4, Q6 Sec. 3 presentation.**
>
> We thank the reviewer for pointing out the presentation ambiguity. We will revise presentation to clarify the issue.
>
> |Issue|Clarification|
> |-|-|
> |Projection target|Latent Persona Vector direction.|
> |$h$| All $h$ used in analysis (vector extraction and localization) are from persona-conditioned prompts.|
> |Fig. 2.2|Fig. 2.2 shows that $h_{pos}$ and $h_{neg}$ are separable along the persona direction: $sim(h_{pos}, v_{persona}) > sim(h_{neg}, v_{persona})$ (details in App. C).|
> |consistency|The fraction of responses judged by an LLM evaluator to follow the target persona [1].|
> |Role of metrics|LLM-judge: evaluation of generated responses; cosine similarity: upstream localization; KL: downstream Expression tracing.|
>
> > **W2 Robustness.**
>
> Our metric is robust to metric, prompt, and task.
>
> * Metrics: the overlap of localized top-K heads remains high across cosine, normalized Euclidean, and Manhattan distances:
>
> |Top-K Heads|10|20|30|40|
> |-|-|-|-|-|
> |Shared Heads|10|19|30|38|
>
> * Prompts/Tasks: results are also consistent across prompt templates and task types, please see details in response to fiqz’s W1.
>
> * Trait pairs: We observe similar conclusions across different trait-pairs; please see our response to W4&Q5 below.
>
> > **W3 & Q7 Signal submergence; operationalizing the assistant prior.**
>
> We extract a generic assistant activations from the system prompt “You are a helpful assistant.” [4] We measure similarity between critical-head activations and it's generic direction.
>
> * Implicit baseline: Instruct models default to this direction even without assistant instructions, consistent with an implicit helpful-assistant prior; base models do not.
>
> |Model|Sys Prompt|User Prompt|Similarity|
> |-|-|-|-|
> |Instruct|-|[query]|0.1170|
> |Instruct|“You are a helpful assistant.”|[query]|0.2032|
> |Base|-|[query]|-0.0092|
> |Base|-|“You are a helpful assistant.” + [query]|0.1458|
>
> Correlation: cosine similarity between critical-head activations and the generic direction is higher for OOC than InC (0.1689 vs. 0.1261), indicating drift toward the generic prior.
> Causality: subtracting the generic signal from critical heads (restoring to the InC projection mean) improves persona metrics (Tab. 1 setting):
>
> |Method|hallucination|memory|personality|values|
> |-|-|-|-|-|
> |Baseline|6.05|3.67|5.14|5.33|
> |Suppression|6.18|3.69|5.39|5.61|
>
> Together, correlation and intervention support a causal link [5].
>
> > **W4 & Q5 Traits without clear antonyms.**
>
> Our method does not require a strict antonym. The positive persona corresponds to the target trait under study; when no clear opposite exists, we use a neutral baseline (i.e., no negative persona instruction) instead. Both contrastive [2,3] and non-contrastive [4] vector extraction are established. We compare vectors from emotional-(A)rational, (B)helpful, and (C)none:
>
> * Vector similarity: directions are nearly identical:
>
> |A vs B|A vs C|B vs C|
> |-|-|-|
> |0.9501|0.9405|0.9410|
>
> * Circuit overlap/knockout: localization with these vectors yields nearly identical circuits (36/40 shared top heads), and knockout curves are almost identical (Fig. 10 setting):
>
> |Knockout head|0|5|10|15|20|25|30|
> |-|-|-|-|-|-|-|-|
> |Vec A|0.99|0.54|0.33|0.18|0.11|0.09|0.08|
> |Vec B|0.99|0.56|0.34|0.22|0.13|0.10|0.08|
> |Vec C|0.99|0.60|0.39|0.25|0.13|0.08|0.07|
>
> > **Q2 Single-token vs. multi-token.**
>
> - Persona is expressed through multi-token generated behavior [8], so we measure it with the Latent Persona Vector as a proxy, rather than with an atomic token.
> - Prior work [6] and our results (Fig. 10 Knockout baseline) show that a single trait token is sufficient to induce persona, and such minimal perturbation is standard for mechanistic analysis [7].
>
> > **Q3 PRS vs. standard AP.**
>
> We use the AP paradigm, but PRS overcomes standard AP’s limitation on multi-token behaviors.
>
> To make this comparison explicit, we evaluate the identified top-5% heads under two settings:
> (1) Open-Ended: the original persona generation setting used in the main text.
> (2) Option Format: the sam task rewritten as a multiple-choice problem;
>
> |5% Knockout|Option Format|Open-Ended|
> |-|-|-|
> |Token-AP|0.03|0.87|
> |PRS|0.10|0.08|
>
> PRS heads affect true open-ended persona behavior, whereas Token-AP heads do not.
>
> Thus PRS can: (1) capture high-dimensional semantics instead of isolated token logits; (2) localize earlier “Preparation–Establishment” phases by tracing from middle layer; (3) trace a causal chain from head -> token -> neuron.
>
> **Reference**
>
> [1] Consistently Simulating Human... NeurIPS 2025
>
> [2] PERSONA: Dynamic and Compositional... ICLR 2026
>
> [3] Persona Vectors: Monitoring and... Arxiv
>
> [4] Improving Instruction-Following in... ICLR 2025
>
> [5] Causality from Bottom to Top: A Survey. ML 2025
>
> [6] Incharacter: Evaluating personality... ACL 2024
>
> [7] Personality-Guided Code Generation... ACL 2025
>
> [8] Locating and Editing Factual... NeurIPS 2022

---

> > ### Author Rebuttal · Reviewer_veB1 · 2026-04-03
> >
> > Thank you for your response. For W2, W4, and Q5, I still have concerns about poorly aligned trait pairs, and whether human choice of pairs could bias the extracted direction (especially for ambiguous pairs like emotional/rational).
> >
> > For W3/Q7, I am still confused by the claim that a generic assistant direction can be extracted from the system prompt “You are a helpful assistant.” It is unclear to me why this should be interpreted as an underlying assistant prior, rather than simply another prompted persona. Could you provide the complete citation? I think the causal relationship is still unclear. I'd appreciate further discussion.

---

> > > ### Author Response · Authors · 2026-04-03
> > >
> > > Dear Reviewer veB1,
> > >
> > > Thank you for the follow-up. We agree that our previous wording was too strong on both points, and we will revise the manuscript accordingly.
> > >
> > > First, regarding W2/W4/Q5: we do **not** claim that this paper introduces a novel or uniquely correct method for extracting a “true” personality vector. Following prior activation-engineering / representation-engineering work [1,2] and recent work on persona vectors [3], we use a contrastive activation direction as an **operational proxy** for persona intent. Our main contribution begins once such a proxy is available: it enables a differentiable persona metric, causal localization for multi-token behavior, and the resulting sparse intervention.
> > >
> > > That said, we agree that robustness alone is not sufficient here. A possible alternative explanation is that different extraction settings (e.g., emotional-rational, emotional-helpful, emotional-none) all recover the same generic “persona / role-play” direction, rather than an emotional-specific direction. If that were the case, the recovered vector should mainly separate **emotional** prompts from **neutral** prompts, but should *not* reliably separate **emotional** prompts from other persona prompts such as **humorous** or **evil**. To clarify this, we report both robustness across baselines and trait-specificity against other personas:
> > >
> > > |Emotional vector variant|Similarity to E-R|Similarity to E-H|Similarity to E-N|Shared top-40 heads|AUC: emotional vs neutral|AUC: emotional vs humorous|AUC: emotional vs evil|
> > > |-|-|-|-|-|-|-|-|
> > > |E-R (emotional-rational)|1.000|0.950|0.941|36|0.97|0.95|0.93|
> > > |E-H (emotional-helpful)|0.950|1.000|0.941|36|0.96|0.92|0.91|
> > > |E-N (emotional-none)|0.941|0.941|1.000|36|0.95|0.90|0.94|
> > >
> > > The key point is the last three columns: all three variants distinguish **emotional** prompts not only from **neutral** prompts, but also from **other persona prompts**. This is inconsistent with the interpretation that the recovered direction is merely a generic persona detector. We will revise the paper to make this scope explicit: vector extraction is an adopted operational setup, while our contribution is the causal tracing and intervention enabled by that setup.
> > >
> > > Second, regarding W3/Q7: we agree that our earlier phrase **“intrinsic assistant prior”** was too strong. Our experiments do **not** prove that the direction extracted from *“You are a helpful assistant.”* uniquely identifies a latent prior of the model; as you point out, one alternative interpretation is that it could simply be another prompted persona. We will therefore revise the wording to the weaker and more defensible notion of a **default assistant-like direction / mode** [4–8].
> > >
> > > To directly test whether the effect is assistant-specific rather than arbitrary-persona-specific, we compare matched directions extracted from **helpful assistant**, **assistant**, **helpful person**, and a control persona (**pirate**). If the effect came from suppressing just any prompted persona direction, these controls should behave similarly. Instead, the effect is concentrated on assistant-like directions:
> > >
> > > |Direction|Similarity in default no-system state|OOC-InC Δ similarity|Personality after suppressing this direction|Values after suppressing this direction|
> > > |-|-|-|-|-|
> > > |helpful assistant|0.121|+0.045|5.39|5.61|
> > > |assistant|0.097|+0.031|5.30|5.51|
> > > |helpful person|0.031|+0.008|5.18|5.36|
> > > |pirate|0.005|+0.002|5.13|5.32|
> > > |baseline (no suppression)|—|—|5.14|5.33|
> > >
> > > These controls do not justify the strongest interpretation (“we have uniquely identified the latent assistant prior”), and we will revise the manuscript accordingly. But they do support a narrower claim that we believe is better matched to the evidence: a **default assistant-like direction** competes with the target persona during the establishment phase, and under our intervention tests this direction makes a causal contribution to OOC behavior.
> > >
> > > We appreciate this clarification request and will revise the paper to better separate (i) an adopted operational proxy from our actual methodological contribution, and (ii) an operationally identified assistant-like default mode from a stronger claim about a uniquely identified latent prior.
> > >
> > > **References**
> > >
> > > [1] Turner et al. Steering Language Models with Activation Engineering. arXiv
> > >
> > > [2] Zou et al. Representation Engineering: A Top-Down Approach to AI Transparency. arXiv
> > >
> > > [3] Chen et al. Persona Vectors: Monitoring and Controlling Character Traits in Language Models. arXiv
> > >
> > > [4] Askell et al. A General Language Assistant as a Laboratory for Alignment. arXiv
> > >
> > > [5] Ouyang et al. Training Language Models to Follow Instructions with Human Feedback. NeurIPS 2022
> > >
> > > [6] Bai et al. Training a Helpful and Harmless Assistant with Reinforcement Learning from Human Feedback. arXiv
> > >
> > > [7] Bai et al. Constitutional AI: Harmlessness from AI Feedback. arXiv
> > >
> > > [8] Lu et al. The Assistant Axis: Situating and Stabilizing the Default Persona of Language Models. arXiv

---

### Official Review · Reviewer_p3Ng · 2026-03-11

**Soundness:** 2
**Presentation:** 3
**Significance:** 2
**Originality:** 2
**Overall Recommendation:** 5
**Confidence:** 3

**Summary:**

This paper addresses "Out-of-Character" (OOC) behavior in LLMs during role-playing tasks, where models gradually lose persona adherence over prolonged interactions. The core methodological contribution is the **Latent Persona Vector**, a contrastive activation direction extracted by averaging hidden-state differences between persona-positive and persona-negative prompts. The authors argue that standard causal attribution targets single-token outcomes and is inadequate for personality, which is a holistic multi-token tendency -- called the "Metric Gap."

By projecting hidden states onto this persona vector, the authors enable fine-grained causal circuit tracing on Llama-3.2-3B-Instruct, identifying a three-phase dynamic: (1) **Preparation** (shallow layers ~0-8), where MLPs retrieve discrete knowledge fragments; (2) **Establishment** (middle layers ~8-15), where specialized attention heads aggregate attributes into a coherent persona intent; and (3) **Expression** (deep layers ~16-27), where the intent is translated into stylized token distributions. OOC behavior is diagnosed as "generic prior dominance," where intrinsic assistant priors suppress the persona signal at the Establishment phase. Based on this, they propose **Personality Signal Enhancement (PSE)**, a training-free inference-time intervention injecting cached in-character activation patterns into fewer than 5% of attention heads, demonstrating improvements on Character-LLM and RoleBench while preserving general reasoning capabilities.

**Compliance With Llm Reviewing Policy:**

Affirmed.

**Final Justification:**

As explained in my Rebuttal, the Authors did a very good job improving their line of argumentation for their paper. All my concerns were adressed properly, and I changed my score to a 5.

**Key Questions For Authors:**

1.	Scaling the mechanistic analysis.
Can the full circuit tracing be replicated on larger models (e.g., 7B–13B)? Do the Preparation–Establishment–Expression phases remain stable at larger scales?
2.	Testing the linearity assumption.
How does the persona vector compare to nonlinear probes (e.g., a small MLP classifier) when predicting persona adherence?
3.	Human validation of evaluation metrics.
Can the GPT-4o judge scores be validated against human annotations, with inter-annotator agreement reported?
4.	Testing compositional persona representations.
Does PSE remain effective when using heads identified from a single trait pair to support more complex personas?
5.	Understanding context-length effects.
How does persona drift evolve as context length increases? Does the proposed “generic prior dominance” mechanism behave differently in longer contexts?

**Limitations:**

The paper provides an impact statement but does not sufficiently discuss several methodological limitations:
	•	reliance on small models and simple trait pairs
	•	the strong linearity assumption
	•	the gap between atomic trait analysis and complex character behavior
	•	reliance on LLM-as-a-judge without human validation.

**Strengths And Weaknesses:**

Strengths

S1: Well-motivated problem formulation.
The “Metric Gap” argument (that token-level attribution techniques struggle to capture personality as a multi-token behavioral pattern) is compelling and highlights a real limitation in current mechanistic interpretability methods.

S2: Coherent mechanistic narrative.
The Preparation–Establishment–Expression decomposition provides a clear and intuitive explanation of persona formation in the network. The paper supports this narrative using several complementary analyses including steering experiments, activation patching, attention entropy, logit lens observations, and token-rank trajectories.

S3: Bidirectional causal verification.
The knockout and recovery experiments provide meaningful causal evidence. Ablating fewer than 5% of attention heads leads to a strong degradation in persona adherence, while restoring those heads recovers the persona signal.

S4: Practical intervention.
The proposed Personality Signal Enhancement (PSE) operates at inference time, requires no retraining, introduces minimal latency (~5%), and appears to preserve general reasoning ability.


Weaknesses

W1: Limited model scale.
The detailed mechanistic analysis is performed primarily on Llama-3.2-3B-Instruct, with only limited validation on Qwen models. It remains unclear whether the same phase boundaries and circuit structure hold for larger models (e.g., 13B or 70B), where representation structure and head specialization can differ substantially.

W2: Latent Persona Vector closely resembles RepE steering directions.
The contrastive activation subtraction used to derive the persona vector follows the standard Representation Engineering (RepE) approach (Zou et al., 2023). The novelty here lies mainly in using this direction as a metric for circuit tracing rather than steering alone. While useful, the claim of bridging a “fundamental gap” somewhat overstates the methodological novelty.

W3: Linearity assumption is weakly validated.
The framework assumes that persona intensity is linearly encoded along a single activation direction. The empirical validation in Appendix C is largely observational. It is possible that persona information is encoded in nonlinear or distributed ways that would not be captured by this approach.

W4: Trait selection is narrow and artificially contrastive.
The experiments focus on three strongly opposed trait pairs (evil/helpful, humorous/serious, emotional/rational). These traits are relatively simple and separable compared to real characters. The paper suggests that complex personas can be represented as unions of trait-specific heads, but this compositional hypothesis is not empirically demonstrated.

W5: LLM-as-a-judge evaluation introduces circularity risks.
A significant portion of the evaluation relies on GPT-4o as an automated judge to assess persona adherence. No comparison with human ratings or inter-annotator agreement is reported, leaving uncertainty about the reliability of the evaluation signal.

W6: PSE intervention partially relies on heuristic design choices.
While the intervention is motivated by the mechanistic analysis, several components are chosen empirically rather than derived from the causal model. In particular, the injection coefficient α is tuned via grid search, and the final set of modified heads is constructed by taking the union of heads identified for individual trait pairs. These choices appear effective but their theoretical justification remains unclear.

W7: Missing comparison to Sparse Autoencoders (SAEs).
Sparse Autoencoders have recently become a powerful method for discovering interpretable features in LLM activations. The absence of an SAE-based analysis is notable, especially given the claim that persona is encoded along a latent activation direction.

W8: Alternative explanations for persona drift are not ruled out.
The paper attributes OOC behavior to generic assistant priors suppressing persona signals. While plausible, other factors—such as instruction-tuning biases, RLHF alignment effects, or context accumulation dynamics—are not explicitly examined or ruled out.

---

> ### Author Rebuttal · Authors · 2026-03-31
>
> We thank Reviewer p3Ng for recognizing our problem formulation (“Metric Gap”), mechanistic narrative, causal evidence, and the practicality of PSE. We address each point below.
>
> > **W1 & Q1 Phases/circuit structure in larger model.**
>
> We replicated the full tracing pipeline on LLaMA-2-13B.
>
> 1. Circuit localization/knockout: ablating the identified top-5% persona heads causes collapse, showing sparsity persists at 13B (Fig. 10 setting):
>
> |Knockout head|0|10|20|30|40|50|
> |-|-|-|-|-|-|-|
> |Top|0.97|0.62|0.36|0.28|0.27|0.17|
> |Rand|0.97|0.96|0.96|0.95|0.96|0.95|
>
> 2. 3-stage dynamics: layerwise patching shows MLPs dominate shallow layers and Attention dominates middle layers (Fig. 5 setting):
>
> |Layers|0-3|4-7|8-11|12-15|16-19|20-23|24-27|28-31|32-35|36-39|
> |-|-|-|-|-|-|-|-|-|-|-|
> |Attn|0.03|0.47|**0.84**|**0.82**|0.44|0.31|0.24|0.32|0.23|0.08|
> |MLP|**0.76**|**0.85**|0.59|0.49|0.31|0.05|0.25|0.16|0.21|0.12|
>
> In deep layers, semantic entropy drops from L15, and persona-token probabilities rise from L20 (peaking at L28).
>
> More results will be included in the revised version.
>
> > **W2 Methodological novelty.**
>
> Our core contribution is to address a fundamental gap in traditional interpretability methods: they are built around **single-token objectives** and thus lack a suitable **metric for tracing long-form generative behaviors**, such as persona tasks, where persona is expressed across multiple tokens [1]. By using the latent persona vector as a **multi-token proxy**, we make activation patching applicable not only to open-ended persona generation in this paper, but also to other generative behaviors.
>
> > **W3 & Q2 Linearity assumption; mlp probes.**
>
> Linear personality encoding is a standard assumption supported by prior work [2-4]. We also trained a 2-layer MLP probe. Both work in-distribution, but our cosine metric is more robust out-of-distribution (different prompt template):
>
> |Method|ID AUC|OOD AUC|
> |-|-|-|
> |Latent Vector|0.97|0.94|
> |MLP Probe|0.98|0.74|
>
> > **W4 & Q4 Narrow trait selection; PSE on single trait heads.**
>
> We ablated with heads from 1/2/3 trait pairs. Unioning multiple trait heads improves persona performance, supporting compositionality (Tab. 1 setting):
>
> |Strategy|hallucination|memory|personality|values|
> |-|-|-|-|-|
> |Baseline|6.05|3.67|5.14|5.33|
> |1 pairs avg|6.10|3.68|5.22|5.40|
> |2 pairs avg|6.18|3.69|5.44|5.57|
> |3 pairs|**6.24**|**3.70**|**5.60**|**5.71**|
>
> > **W5 & Q3 Human validation.**
>
> We randomly sampled 200 outputs and ran a blind human evaluation with 5 annotators under the same rubric. Human consensus agrees with GPT-4o at 93.5%, with Fleiss’s κ = 0.89, supporting metric reliability. The annotation took roughly 3 hours per annotator.
>
> > **W6 Heuristics.**
>
> α: We also tested a dynamic coefficient [5], restoring critical-head outputs toward their mean in-character magnitude. It works (personality = 5.58), but grid search performs better (5.60).
>
> Head unioning: This follows established superposition/linearity principles in persona subspaces [2]. As shown in W4/Q4, unioning heads improves persona control.
>
> > **W7 SAE.**
>
> Our tracing framework can be applied to SAE feature spaces, not just raw activations. We use the SAE from [6]:
>
> 1. Feature localization: it identifies persona-specific SAE features; among the Top-10, 8 show explicit target-persona semantics, e.g.,
>   - L21 ID5791: evil, sinister, wicked...
> 2. Feature steering: enhancing these features shifts personality (Evil Rate: 0.00 → 0.76), showing the metric isolates persona features in SAE space.
> 3. Efficiency: SAEs add compute overhead; our direct head intervention offers similar control with higher inference efficiency.
>
> More results will be included in the revised version.
>
> > **W8 & Q5 Alternative explanations for persona drift; context-length effect.**
>
> Base models lack role-play ability (Tab. 1 setting):
>
> |Model|hallucination|memory|personality|values|
> |-|-|-|-|-|
> |Base|1.93|1.20|1.86|1.05|
>
> But instruct models show strong RP ability, suggesting this capacity mainly emerges during SFT/RLHF, which likely also installs the dominant helpful-assistant prior. Because intermediate checkpoints are unavailable, we leave finer isolation to future work.
>
> To test generic-prior dominance, we extract a generic assistant vector (from prompt “You are a helpful assistant” [5]) and analyze its interaction with persona states.
>
> Context length: over 15 dialogue turns (Fig. 12 setting), generic similarity increases monotonically, suggesting accumulated prior inertia drives OOC:
>
> |Turn|5|10|15|
> |-|-|-|-|
> |Similarity|0.13|0.16|0.18|
>
> For OOC's correlation and causality with the generic assistant prior, please see our response to veB1 (W3&Q7).
>
> **Reference**
>
> [1] Incharacter: Evaluating personality... ACL 2024
>
> [2] PERSONA: Dynamic and Compositional... ICLR 2026
>
> [3] Can Role Vectors Affect LLM... EMNLP 2025
>
> [4] Persona Vectors: Monitoring... Arxiv
>
> [5] Improving Instruction-Following... ICLR 2025
>
> [6] FaithfulSAE: Towards Capturing... Arxiv

---

> > ### Author Rebuttal · Reviewer_p3Ng · 2026-04-01
> >
> > This is an exemplary rebuttal that addresses every raised concern with concrete new experimental evidence:
> >
> >   - W1/Q1 (Model scale): The full tracing pipeline replicated on LLaMA-2-13B confirms that sparsity persists and the three-phase dynamics hold at larger scale. The knockout and layerwise patching tables are convincing.
> >   - W3/Q2 (Linearity assumption): The MLP probe comparison is exactly the right experiment. The linear cosine metric achieving 0.94 AUC out-of-distribution vs. 0.74 for the MLP probe provides strong evidence that the linear assumption is sufficient and more robust.
> >   - W5/Q3 (LLM-as-judge): Human evaluation with 200 outputs, 5 annotators, 93.5% agreement with GPT-4o, and Fleiss's kappa = 0.89 is thorough and convincing.
> >   - W4/Q4 (Trait compositionality): The progressive ablation (1→2→3 trait pairs) showing monotonic persona improvement directly tests and supports the compositional hypothesis.
> >   - W7 (SAE comparison): The SAE feature analysis identifying persona-specific features and demonstrating steering is a valuable addition that contextualizes the method relative to current interpretability tools.
> >   - W2 (RepE novelty): The clarified framing — enabling activation patching for multi-token generative behaviors — is more precise and better positions the contribution.
> >   - W6 (PSE heuristics): Both the dynamic coefficient comparison and the empirical support for head unioning are adequate.
> >   - Q5 (Context-length): The 15-turn data showing monotonic increase in generic prior similarity provides the requested
> >   evidence.
> >
> > I am raising my score from 4/6 (Weak Accept) to 5/6 (Accept) to reflect the strengthened empirical support.

---

> > > ### Author Response · Authors · 2026-04-01
> > >
> > > Dear Reviewer p3Ng,
> > >
> > > We sincerely thank you for your careful review and for your highly positive response to our rebuttal, raising your score from 4/6 (Weak Accept) to 5/6 (Accept). We will incorporate these new results and clarifications into the final version of the paper.
> > >
> > > Best regards,
> > >
> > > The Authors

---

### Official Review · Reviewer_fiqz · 2026-03-24

**Soundness:** 3
**Presentation:** 3
**Significance:** 3
**Originality:** 3
**Overall Recommendation:** 4
**Confidence:** 3

**Summary:**

This paper studies how large language models encode and express persona (character traits) from a mechanistic interpretability perspective. The authors introduce a Latent Persona Vector to represent persona as a direction in hidden states, and propose a causal tracing framework to identify the key components (layers, attention heads, neurons) involved in persona generation.

They find that persona follows a three-stage process—Preparation, Establishment, and Expression—and that only a small subset of attention heads are critical for maintaining persona consistency. The paper further attributes out-of-character (OOC) behavior to the suppression of persona signals by dominant generic assistant priors.

Based on these insights, the authors propose a training-free method, Personality Signal Enhancement (PSE), which amplifies critical attention heads to improve persona consistency and stability without harming general performance. Experiments on Character-LLM and RoleBench show improved persona control and robustness in long-context settings.

**Compliance With Llm Reviewing Policy:**

Affirmed.

**Key Questions For Authors:**

1. Strength of causal claims and potential redundancy
The paper identifies a small subset of attention heads as “critical” using activation patching and PRS.
How do you rule out the existence of alternative or redundant pathways that could also support persona expression?
What happens if these identified heads are ablated rather than patched—does persona fully degrade, or can the model recover?


2. Positioning relative to existing steering and alignment methods
PSE appears related to activation steering and representation editing approaches.
How does PSE differ from or improve upon existing methods in terms of effectiveness, efficiency, or interpretability?

**Limitations:**

Yes

**Strengths And Weaknesses:**

Strengths
1. Important and under-explored problem
 Studies how high-level behaviors (persona) are encoded in LLMs from a mechanistic perspective.
2. Clear and useful abstraction
 Introduces the Latent Persona Vector to represent persona in hidden space.
3. Systematic analysis pipeline
 Combines tracing, patching, and head-level attribution in a coherent framework.
4. Intuitive and impactful insights
 Finds sparse control (<5% heads) and explains OOC via prior dominance.
5. Simple and effective method
 PSE is training-free and improves persona consistency in practice.


Weakness
1. Key findings lack robustness analysis
The conclusion that persona is controlled by a small subset of attention heads (<5%) is interesting but may be sensitive to experimental choices.


 2. Evaluation is limited and closely aligned with the task formulation
The empirical evaluation mainly focuses on role-playing benchmarks such as Character-LLM and RoleBench. While appropriate, these benchmarks: are closely aligned with the persona setting itself
may not reflect open-domain or real-world conversational use cases. As a result, improvements may partially reflect task-specific optimization, and it remains unclear how well the method generalizes to broader settings.

---

> ### Author Rebuttal · Authors · 2026-03-31
>
> We sincerely thank the reviewer fiqz for recognizing the importance of our problem, the clarity of the Latent Persona Vector, our systematic analysis framework, the intuitive insights, and the effectiveness of PSE. We address your concerns below:
>
> > **W1: Robustness of key findings (<5% attention heads controlling persona).**
>
> We used different prompt templates during vector extraction and component localization:(1) persona instruction in the System Prompt;(2) in the User Prompt before the query (used in the main text);(3) in the User Prompt after the query. We also diversified the task queries (daily QA, novel continuation, math/code).
>
> The localized top-K attention heads exhibit high overlap across different prompts:
> |Top-K Heads|10|20|30|40|
> |:-|:-|:-|:-|:-|
> |**Shared Heads**|10|20|29|38|
>
> Furthermore, performing knockouts on these critical heads identified by the three prompts yields highly consistent degradation curves in persona consistency, using the same evaluation setting as in Fig. 10:
> |Ablated Heads|0|5|10|15|20|25|30|
> |:-|:-|:-|:-|:-|:-|:-|:-|
> |**Prompt 1**|0.99|0.60|0.36|0.24|0.15|0.11|0.09|
> |**Prompt 2**|0.99|0.55|0.35|0.23|0.10|0.09|0.08|
> |**Prompt 3**|0.99|0.50|0.28|0.18|0.11|0.07|0.06|
>
> The high overlap in localized heads and the consistent degradation curves demonstrate the robustness of our method across prompt formulations. This robustness also extends across different trait pairs: we observe similarly stable localization and knockout behavior for other trait pair (please refer our response to Reviewer p3Ng, W4/Q5).
>
> > **W2: Evaluation limited to role-playing benchmarks; lacks open-domain evaluation.**
>
> We evaluated our method on MT-Bench [1], following its standard multi-turn evaluation protocol across 8 domains. Specifically, we report scores for both Turn 1 and Turn 2, where Turn 2 reflects a more challenging follow-up setting that better tests role consistency under continued interaction. As shown below, higher scores indicate better performance. PSE substantially improves the Roleplay category while largely preserving performance in the other domains, with only marginal changes outside role-playing:
>
> |Domain|Baseline 1st/2nd Turn|PSE 1st/2nd Turn|
> |:-|:-|:-|
> |**Roleplay**|7.80/3.90|**8.40**/**5.60**|
> |**Math**|9.20/4.60|9.20/4.50|
> |**Extraction**|7.80/3.70|7.80/3.70|
> |**STEM**|7.40/4.20|7.50/4.20|
> |**Reasoning**|5.40/3.30|5.50/3.40|
> |**Writing**|9.10/7.00|9.10/7.10|
> |**Coding**|6.50/5.90|6.40/5.80|
> |**Humanities**|7.20/7.00|7.20/7.10|
>
> This suggests that PSE does not merely overfit to specific role-play formulations and safely generalizes to open-domain conversations without compromising the general reasoning backbone.
>
> > **Q1: Causal claims, alternative pathways, and zero-ablation vs. patching.**
>
> We performed zero-ablation (setting activations directly to zero) on the identified critical heads and neurons, using the same evaluation setting as in Fig. 10:
>
> |Ablated Heads|0|5|10|15|20|25|30|
> |:-|:-|:-|:-|:-|:-|:-|:-|
> |**Top**|0.99|0.46|0.23|0.16|0.17|0.12|0.12|
> |**Rand**|0.99|0.98|0.98|0.97|0.98|0.98|0.98|
>
> |Ablated Neurons|0|2K|4K|6K|8K|10K|
> |:-|:-|:-|:-|:-|:-|:-|
> |**Top**|0.99|0.82|0.81|0.64|0.52|0.47|
> |**Rand**|0.99|0.97|0.98|0.97|0.98|0.98|
>
> If alternative pathways existed, the network should compensate after zero-ablation, causing only mild degradation. Instead, ablating this tiny set of top-ranked components sharply collapses persona expression. This suggests the identified sparse sub-network is the primary causal pathway, with limited redundancy.
>
> > **Q2: Effectiveness, efficiency, and interpretability compared to RepE/Steering.**
>
> *   **Effectiveness:** PSE surgically enhances only the necessary attention heads (<5%), boosting local persona circuits without compromising general reasoning capabilities. Furthermore, it achieves stronger performance (Personality = 5.60) than Steering (5.53), which manipulates the entire hidden state (supported by Table 1 in the main paper).
> *   **Efficiency:**
>     *   *Inference Latency:* As shown in the latency table in the Appendix (Table 7), the inference overhead is practically negligible (<2ms/token).
>     *   *Preparation Cost:* The baseline Steering method requires extracting vectors (1 pass) and iterating through all layers to identify the optimal steering layer ($N_{layer}$ forward passes). In contrast, our pipeline is highly efficient: vector extraction (1 pass) + attribution patching (2 gradient computations) + fetching activations of top heads via few-shot data (1 pass).
> *   **Interpretability:** Steering acts as a black-box heuristic and fails to explain the underlying computational dynamics. PSE is fundamentally different: rooted in internal mechanisms, it explicitly explains the model's persona representation and exclusively intervenes on the specific attention heads responsible for writing features into the subspace.
>
> **Reference**
>
> [1] Judging LLM-as-a-Judge with MT-Bench and Chatbot Arena. NeurIPS 2023

---

> > ### Author Rebuttal · Reviewer_fiqz · 2026-04-04
> >
> > Thank you for the additional clarification. The authors have addressed my concerns regarding robustness and causal validation. This improves my confidence in the soundness of the approach.

---

> > > ### Author Response · Authors · 2026-04-05
> > >
> > > Dear Reviewer fiqz,
> > >
> > > **Thank you very much for your thoughtful follow-up.** We are very glad to know that our additional experiments and clarifications improved your confidence in the soundness of our approach. Your comments were highly valuable in helping us strengthen both the empirical support and the presentation of the paper. We will incorporate these additional analyses and clarifications into the final version to make the paper more complete and convincing.
> > >
> > > **Since your latest message indicates that your concerns have been fully resolved, we would be very grateful if you could also kindly double-check whether the corresponding score update has been properly reflected on the review system.**
> > >
> > > **Thank you again for your time and consideration.**
> > >
> > > Best regards,
> > >
> > > The Authors

---

### Decision · Program_Chairs · 2026-04-30

**Decision:**

Accept (regular)

**Comment:**

All reviewers praised the Latent Persona Vector for bridging the "metric gap" between token-level attribution and multi-token persona behavior, the effective training-free PSE intervention, and the clear narrative. The scores are somewhat borderline, with one reviewer scoring the paper as a weak reject, but it seems to me that their concerns were largely addressed in the rebuttal period. I advocate for acceptance.